# A probabilistic model for fracture events of Petermann ice islands under the influence of atmospheric and oceanic conditions

Reza Zeinali-Torbati[1], Ian D. Turnbull[2], Rocky S. Taylor[1], Derek Mueller[3]

[1]Faculty of Engineering and Applied Science, Memorial University of Newfoundland, St. John's, NL A1B 3X5, Canada
[2]Ice Engineering, C-CORE, St. John's, NL A1B 3X5, Canada
[3]Department of Geography and Environmental Studies, Carleton University, Ottawa, ON K1S 5B6, Canada

*Correspondence to*: Reza Zeinali-Torbati (rzt313@mun.ca)

**Abstract.** Four calving events of Petermann Glacier happened in 2008, 2010, 2011, and 2012, which resulted in the drift and deterioration of numerous ice islands, some reaching as far as offshore Newfoundland. The presence of these ice islands in the eastern Canadian Arctic increases the risk of interaction with offshore operations and shipping activities. This study uses the recently developed Canadian Ice Island Drift, Deterioration and Detection database to investigate the fracture events that these ice islands experienced, and presents a probabilistic model for the conditional occurrence of such events by analyzing the atmospheric and oceanic conditions that drive the causes behind the ice island fracture events. Variables representing the atmospheric and oceanic conditions that the ice islands were subjected to are extracted from reanalysis datasets and then interpolated to evaluate their distributions for both fracture and non-fracture events. The probability of fracture event occurrence for different combinations of input variable conditions are quantified using Bayes' theorem. Out of the seven variables analyzed in this study, water temperature and ocean current speed are identified as the most and least important contributors, respectively, to the fracture events of the Petermann ice islands. It is also revealed that the ice island fracture probability increases to 75% as the ice islands encounter extreme (very high) atmospheric and oceanic conditions. A validation scheme is presented using cross-validation approach and Pareto principle, and an average error of 13-39% is reported in the fracture probability estimations. The presented probabilistic model has a predictive capability for future fracture events of ice islands and could be of particular interest to offshore and marine ice/risk management in the eastern Canadian Arctic. Future research, however, is necessary for model training and testing to further validate this ice island fracture model.

## 1 Introduction

With the advancement of offshore operations and shipping activities into the harsh environment in the eastern Canadian waters, these activities are being subjected to greater risks from glacial ice features (Saper, 2011). The shipping and resource extraction industries in this region, therefore, require a better understanding of the dynamics and physical properties of these ice features to be able to devise appropriate ice management strategies for safe operations. Specifically, a better understanding of the drift and deterioration characteristics of icebergs and ice islands (large tabular icebergs) is needed for risk management strategies. However, due to the occasional presence of ice islands in regions with lower latitudes such as offshore Newfoundland and

Labrador (Johannessen et al., 2011), there has been limited research on the dynamics of ice islands, when compared to the research concerning annual presence of smaller icebergs in the region. Ice island research studies have been mainly focused on their potential risks to shipping activities and offshore operations (Peterson, 2011; Mueller et al., 2013; Fuglem and Jordaan, 2017), as well as their meltwater input as they deteriorate and melt over large regions (Stern et al., 2015; Merino et al., 2016; Wagner et al., 2017; Crawford et al., 2018d).

For offshore operations, whether or not a drifting ice feature is manageable, or if activities should be suspended, depends on its velocity, mass, and the number of ice features approaching the vicinity. Mass and size distribution are both dependent on the deterioration of each ice feature via melting and/or fracturing, processes that are associated with several atmospheric and oceanic variables. Therefore, it is important to evaluate the effect of metocean (meteorology and oceanography) variables on the deterioration of ice islands. However, the harsh conditions in the eastern Canadian marine environment make it very challenging to collect in-situ metocean data. Also, due to the difficulties involved in tracking a large number of ice islands over their lives (Merino et al., 2016; Rackow et al., 2017; Stern et al., 2016), it has been a challenge to investigate the conditions that lead to ice island deterioration. An alternative way to track a large number of ice islands and monitor their deterioration is using remote sensing observations. Unlike optical methods, Synthetic Aperture Radar (SAR) imagery can produce images of ice islands even when the amount of daylight is low and cloud cover is high (Jeffries, 2002). The recent calving events of Petermann Glacier (2008-2012) and other northern Greenland glaciers have been accompanied by SAR satellite imagery (July 2008 to December 2013), which allowed tracking the ice islands throughout their periods of drift and deterioration. Through a collaboration between the Canadian Ice Service (CIS) and Water and Ice Research Lab (WIRL) at Carleton University, a large number of SAR images from the CIS archive were analyzed using a geographical information system to develop a geospatial database associated with the ice islands originally calved from Petermann Glacier in 2008-2012, as well as the Ryder, Steensby, C.H. Ostenfeld, and North Greenland ice tongues (Crawford et al., 2018a). The ice islands were delineated and monitored as they deteriorated (via melting and/or fracturing) down to a threshold of 0.25 km$^2$ in surface area (Crawford et al., 2018d), and the information was recorded in the Canadian Ice Island Drift, Deterioration and Detection (CI2D3) database. The calving events of the Petermann Glacier in 2008, 2010, 2011, and 2012 corresponded to the removal of 36, 302, 4, and 145 km$^2$ from the Petermann Ice Tongue, respectively (Crawford et al., 2018a). The calving event that occurred in 2010 was the most significant of all and resulted in the loss of about 25% of the Petermann Glacier ice tongue (Nick et al., 2012). These calving events generated numerous smaller ice islands that drifted southwards toward the Labrador Sea, which were tracked in the CI2D3 database. More information on the CI2D3 database and its documentation can be found in Desjardins et al. (2018) and Crawford et al. (2018b).

## 1.1 Past studies on iceberg deterioration

A key component for a reliable ice drift model and risk assessment of icebergs is the ability to estimate their mass (Crawford et al., 2018c), a variable that constantly changes as a result of melting and small/large scale fracturing as it drifts. This can be investigated through melt rate and small-scale physical calving models, field measurements, or remote sensing observations.

Iceberg melt rate models predict processes such as forced convection caused by air at the iceberg sail and water at the keel, solar radiation on the iceberg sail, natural convection on sidewalls, and sidewall erosion at the waterline caused by waves (Job, 1978; El-Tahan et al., 1987; Savage, 2001). The resulting iceberg calving caused by wave erosion has been modeled using empirical models (*e.g.*, White et al., 1980; Savage, 2001) and physical models (*e.g.*, Wagner et al., 2014). Savage (2001) studied the relative contribution of each of these mechanisms to the overall deterioration of three different icebergs and found

that wave erosion at the waterline was the dominant mechanism in contributing to the overall iceberg deterioration (by 50-65%), followed by the resultant wave-induced calving events (by 20-30%). Savage (2001) also found that surface melt played a minor role in the deterioration of icebergs, but it was revealed in another study that the deterioration of ice islands was significantly influenced by the surface melt due to the large surface area of ice islands (Crocker et al., 2013). Kubat et al. (2007) used the deterioration mechanisms described by Savage (2001) to build an operational iceberg forecasting model for

the CIS. The sensitivity of the deterioration model to various metocean variables was examined, and it was revealed that the overall deterioration of icebergs was most significantly influenced by wave height (via erosion at the waterline and calving of the overhanging slabs), followed by water temperature (Kubat et al., 2007). The importance of waves and wave-related calving in the overall deterioration of icebergs was also highlighted by Rackow et al. (2017) who investigated the influence of the wave-induced calving, basal melt, and buoyant convection on the deterioration of 6912 icebergs with varying sizes (0.3-4717.6

80    km$^2$) in the Antarctic. Rackow et al., 2017 highlighted the importance of iceberg size in their thermodynamic characteristics and that while waves played the most important role in the decay of smaller icebergs (<10 km), basal melt was an important contributor to the overall mass loss of giant icebergs (>10 km). In a similar study, Stern et al. (2017) presented a novel framework to simulate drifting tabular icebergs for climate studies. The authors modeled the melt of tabular icebergs submerged in the ocean in the Antarctic through the three mechanisms that were used in Rackow et al. (2017). Crawford et al.

(2015) modeled the energy fluxes at the surface using the bulk aerodynamic approach to estimate the surface melt of an ice island sail, and validated the results against three surface ablation models (from Kubat et al., 2007; Ballicater Consulting Ltd., 2012; Hock, 2003). Bouhier et al. (2018) studied the observed vertical melt of two large Antarctic icebergs through the combined analysis of satellite altimetry and imagery and compared this against melt rate estimates from two different models: a forced convection approach and a thermal turbulent exchange approach. While the former approach was found to

underestimate the iceberg melt rates, the latter approach was more reliable in modeling iceberg thickness variations. Zeinali-Torbati et al. (2020) estimated the reduction in the mass of four ice island fragments offshore Newfoundland using a surface ablation model (from Hock, 2003), a basal ablation model (from Crawford, 2018), and the observed areal surface reduction from SAR images. The authors revealed that basal melt had a greater contribution to the overall thickness melt of the ice island fragments, which was in agreement with the results of the thickness melt model in Crawford (2018) and Bouhier et al. (2018).

In a field study (Halliday et al., 2012), however, the melt and thinning rate of an ice island offshore Labrador was measured using ablation stakes and ground penetrating radar, and it was revealed that surface ablation contributed more than basal ablation to the overall thinning rate of the ice island. In other fieldwork, Crawford et al. (2020) deployed an ice-penetrating radar on an ice island originated from the 2012 calving event of Petermann Glacier to measure the surface and basal ablation

rates over an 11-month period. It was revealed that while basal ablation contributed to 73% of the total thinning rate, it played a minimal role in the ice island overall mass loss when compared to areal surface reduction likely caused by wave erosion, wave-induced calving, and fracture events. The authors, however, stated that basal ablation significantly influences the thickness of ice islands, which would likely increase the probability of large-scale fracture event occurrence (Crawford et al., 2020).

The deterioration mechanisms mentioned earlier describe formulations for the melt rates and small-scale calving events of icebergs caused by various metocean conditions. However, there are other mechanisms associated with iceberg deterioration such as large-scale fracture caused by internal stress and convection caused by iceberg rolling (Kubat et al., 2007). While fracture mechanisms play a more important role than melting in the overall deterioration of large icebergs (Bouhier et al., 2018), they are not as well studied and are often neglected due to the infrequent occurrence of fracture events (Kubat et al., 2007). It has also been difficult to model these processes using physical models due to the lack of quantitative theories to explain these mechanisms (Savage, 2001). To date, there are only a few deterministic models to describe the large-scale fracture mechanisms for icebergs (*e.g.*, Diemand et al., 1987; Wagner et al., 2014; Bouhier et al., 2018; England et al., 2020). Additionally, the accuracy of deterministic deterioration models for glacial ice will be limited by the uncertainty in the physical parameters that govern the deterioration processes. Iceberg fracture processes have previously been studied using numerical methods to investigate fracture events for different iceberg geometries (Bassis and Jacobs, 2013), due to buoyancy-driven flexure (Sazidy et al., 2019), as well as due to the accumulation of microcracks in the ice structure (Bahr, 1995). Also, a recent study (Smith, 2020) investigated the fracture events of ice islands when a large protuberance develops on their keels, where a finite element analysis was used to estimate the buoyancy-driven bending stress and predict the associated fractures. However, these numerical models did not account for the relative role of metocean conditions in the fracture processes. Probabilistic methods, however, have the ability to account for the relative contribution of meteorological and hydrological conditions to the fracture events of glacial ice features. Bouhier et al. (2018) investigated the fracture-related decay of two large Antarctic icebergs through analyzing the correlation between their relative volume loss and environmental variables (sea surface temperature, current speed, difference of iceberg and current velocities, significant wave height, wave peak frequency, and wave energy at the bobbing period). The authors found that while wave-related quantities had no significant impact on the relative volume loss, sea surface temperature and iceberg velocity showed the highest correlation with the observed volume loss. Based on these two salient variables, Bouhier et al. (2018) characterized fracture events using a probability distribution and presented a deterministic bulk fracture model, which performed successfully in the estimation of iceberg relative volume loss. However, they noted that given the stochastic nature of fracturing process, individual fracture events cannot be predicted. England et al. (2020) presented an approach for modelling the fracture events of large tabular icebergs by incorporating a stochastic representation of the "footloose mechanism" (Wagner et al., 2014) into the analytical iceberg drift by Wagner et al. (2017). The authors showed that coupling their fracture model with an analytical drift model significantly impacted the iceberg meltwater distribution and resulted in improved simulated iceberg trajectories. England et al. (2020), however, noted that the fracture mechanism in their model is simplified based on several assumptions, a key one being the probability of a child iceberg

fracturing from the parent iceberg is set as constant in time. However, this parameter should be, in fact, dependent on the environmental variables such as sea surface temperature.

The fracture models noted above are not able to quantify the probability of fracture events under different atmospheric and oceanic conditions, a quantity that can be estimated using Bayesian approach. To date, no previous research has adopted Bayesian approach to predict the probability of ice island fracture events under the influence of the metocean conditions that control these events, likely due to the lack of reliable data. However, several studies have adopted a probabilistic approach using a Bayesian belief network and hydro-meteorological variables for navigational risk assessment of ships (Zhang et al.,

2013) or to estimate the conditional probability of ship besetting in sea ice covered waters (Turnbull et al., 2019; Fu et al., 2016; Montewka et al., 2015; Montewka et al., 2013). This study uses the CI2D3 database and adopts a similar methodology to that used in these besetting studies to present a probabilistic fracture model for ice islands as a function of the metocean conditions.

       The CI2D3 database was previously used by Crawford et al. (2018d) to investigate the size distributions and meltwater

flux of Petermann ice islands. The analysis of size distribution revealed that small ice islands constituted a significant proportion of ice island population, but large ice islands contributed the most to the total mass and melt water flux (Crawford et al., 2018d). The authors also revealed that fracture processes significantly contributed to the overall deterioration of Petermann ice islands as the ice island size distribution followed a power law model, which corroborated the results of Stern et al. (2016), Tournadre et al., (2016), Enderlin et al. (2016), Bouhier et al. (2018), and Barbat et al. (2019).

This study uses the CI2D3 database to study fracture events of the ice islands which originated from the calving events of Petermann Glacier in 2008-2012. Various atmospheric and oceanic variables are analyzed to probabilistically determine the set of conditions that lead to the highest chance of fracture event occurrence. This study first presents a description of the data structure in the CI2D3 database. Then, an overview of the results from a preliminary data analysis on the deterioration of Petermann ice islands is presented, followed by the results of the probabilistic fracture model. Finally, a

validation scheme is presented to quantify the accuracy of the probabilistic fracture model.

## 2 Methodology

### 2.1 Data extraction from CI2D3 database

The CI2D3 database (version 1.1) contains data extracted from around 25,000 satellite imagery observations of ice islands from various glaciers, including the Petermann, Ryder, Steensby, C.H. Ostenfeld, and North Greenland glaciers. The data

contains a geospatial polygon and 28 attribute fields for each observation. An algorithm was developed in MATLAB (version R2017b) to extract the data subsets associated with the 2008, 2010, 2011, and 2012 Petermann ice islands (17,755 observations). For each observation, the spatial and temporal data (latitude, longitude, and time) were extracted. Here it should be noted that the "birth" or beginning of a given ice island is considered to be immediately after it calved from another ice island (or glacier), and the "death" or end of that feature is taken as when it calves into two or more fragments. By this

definition, 328 fracture events and 845 ice islands were identified. The ice islands were tracked in the CI2D3 database, and the parent-child relationship between the ice islands was captured as fracture events happened. To identify the parent-child relationship, the unique identifier for each ice island observation was extracted and matched with the *lineage* and *mother* fields (*i.e.*, fields in the database structure that tie subsequent observations together and relate the ice islands to their parents). This permits identification of the previous observations of each ice island back to the time it was born, which were later used for

estimating the cumulative effect of variables (*e.g.*, air and water temperatures, and waves) that each ice island experienced over its lifespan. The algorithm also used the *ddinfo* field to identify if the ice island was grounded or drifting at the time of observation. This was used to estimate the grounding time over the lifespan of each ice island.

## 2.2 Atmospheric and oceanic data extraction

A series of atmospheric and oceanic data were collected from reanalysis databases in the region of interest between northwest

Greenland and offshore Newfoundland (46-83 °N, 45-95 °W) from July 2008 to December 2013. Daily average values (0.3° spatial resolution) for zonal-meridional components of 10-m wind velocity ($m\ s^{-1}$) and 2-m air temperature (℃) were extracted from the North American Regional Reanalysis (NARR); daily average values (1/12°) for zonal-meridional components of ocean current velocity ($m\ s^{-1}$) and potential water temperature (℃) in 25 depth layers (down to 156 m) from the Global Ocean Physics Reanalysis model in Copernicus Marine Environment Monitoring Service (CMEMS); six-hourly

values (1/8°) for significant height of combined wind waves and swell ($m$) and mean wave period ($s$) from the European Centre for Medium-Range Weather Forecasts (ECMWF) ERA-Interim Reanalysis; and sea ice concentration (%) from the CIS digital daily ice charts. The extracted metocean data were linearly interpolated in space and time to the positions and times of the ice island centroids recorded in the CI2D3 database to represent the distribution of atmospheric and oceanic conditions over the drift tracks of the ice islands.

The temporal resolution of the satellite observations in the CI2D3 database were not consistent for all ice islands over their drift periods. The reanalysis variables used in this study were usually available sub-daily (hourly, three-hourly, or six-hourly), but the temporal resolution of the images used to create the CI2D3 database range periodically from sub-daily to bi-weekly. Therefore, all atmospheric and oceanic data were extracted or averaged as daily values and then interpolated in space and time to the positions and times of the ice island observations.

## 2.3 Probabilistic model development

To evaluate the conditional dependence of ice island fracture events on atmospheric and oceanic variables, a Bayesian approach was employed. Bayesian analysis is a well-used method in probabilistic studies to evaluate the probability of a certain outcome using the most salient predictive variables (Gutierrez et al., 2011). This method is recognized by its strong reasoning ability in uncertain situations and its ability to combine and analyze data from various datasets (Fu et al., 2016). Using the Bayesian

method, the dependency and independency of a set of variables (Fig. 1-a) were analyzed via a directed acyclic graph (Fig. 1-b). Initially, the distribution of 10 atmospheric and oceanic variables were studied. These included wind speed ($V_w$), air

temperature ($T_a$), ocean current speed ($V_c$), water temperature ($T_w$), wave energy index ($E_w$), lifetime mean air temperature ($T_{a\_avg}$), lifetime mean water temperature ($T_{a\_avg}$), lifetime mean wave energy index ($E_{w\_avg}$), grounding time ($t_g$), and sea ice concentration ($C_{si}$). Wind and current speeds were estimated as the magnitude of the extracted zonal-meridional components for wind and ocean currents, respectively. Water temperature was estimated as the average of water temperatures at all layers from the water surface down to a depth of 50 m. Given that wave energy flux is proportional to mean wave period and significant wave height squared (Christakos et al., 2020; Akpınar et al., 2019; Waters, 2008; Falnes, 2007), the wave energy index was defined as:

$$E_w = (H_{wave})^2 (T_{wave}), \qquad\qquad\qquad (1)$$

where $H_{wave}$ represents the significant height of combined wind waves and swell ($m$) and $T_{wave}$ represents the mean wave period ($s$). Time since previous calving was also explored as a variable, but this had very little predictive power and was excluded from subsequent analyses. However, in order to capture the cumulative effect of temperature and waves over the lifespan of the ice island, we first identified all previous observations of each ice island back to the time and location that it was born. Then, the air and water temperatures, and wave energy index were interpolated spatially and temporally to all previous observations. To exclude the effect of observation frequency, these variables were interpolated in space and time to a daily interval. These daily-average values were then averaged over the number of days the ice island drifted to effectively compute the lifetime mean wave energy index, as well as the mean air and water temperatures over the lifespan of the ice island, which differ from positive degree day calculations that are often used in ice melt rate models (*e.g.*, Hock, 2003). To better capture the short-lived extreme conditions in air and water temperatures prior to fracture events, the two-week mean values for air and water temperatures were also tested, but they did not make significant difference in the outcome and brought no improvement to the model performance, so they were subsequently excluded from the model inputs. Grounding time was estimated by adding the number of days that an ice island was grounded over its lifespan. The distribution of these variables around fractured ice islands and all ice island observations were investigated and compared. However, further analysis of $E_{w\_avg}$ and $t_g$ distributions for fracture events and all observations identified no correlation between these variables and the occurrence of fracture events, so they were excluded from the Bayesian fracture model development. Also, studying the sea ice cover around the ice island observations showed that sea ice concentrations were less than 3/10ths for about 99% of the observations, so it was discarded from further analysis as a model input variable. Sea ice, however, may play a role in fracture events of ice islands in other regions (*e.g.*, England et al., 2020), so the presented model would need to be extended for application in such regions.

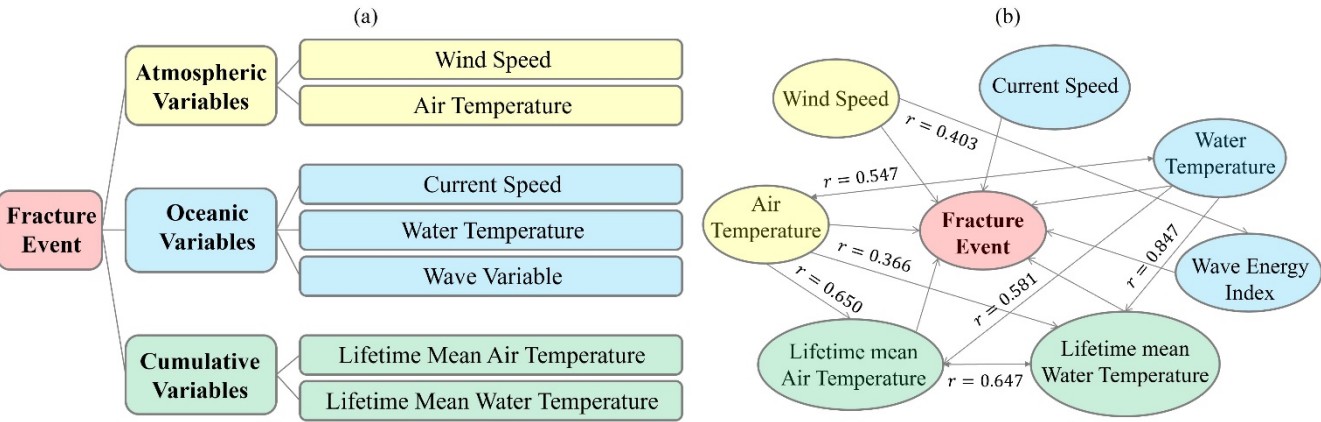

**Figure 1.** The classification of variables for the Bayesian fracture network (a), and the associated directed acyclic graph that shows the inter-relationship between the variables with the arrowheads showing the causality and the r values showing the associated correlation coefficients (b)

The distribution of the metocean variables presented in Fig. 1 were studied for all ice island observations and compared against the variable distributions at the time of fracture events. However, the extracted reanalysis data revealed different number of data points available for the analysis of the seven variables presented. For example, for wind speed and air temperature, 17735 data points were available for all observations of Petermann ice islands. For current speed and water temperature, 16791 and 16784 data points were available for all observations of Petermann ice islands, respectively. These points covered most of the spatial and temporal records from the CI2D3 database. However, for wave energy index, only 3985 data points from all observations were available during the same time period. Similarly, different number of data points were available for each variable during the fracture events. The different numbers of data points for each variable are likely due to the fact that the ice islands drifted near the coastlines at times, and these data were extracted from reanalysis models that have insufficient spatial resolution to model data close to the coastlines. Therefore, the distributions of the studied variables from all observations and fractured subset are represented by relative frequencies to allow for consistent comparison of these distributions.

The correlation between a pair of variables was investigated using the Pearson Product-Moment Correlation coefficient ($r$), given by (Freedman et al., 2010):

$$r = \frac{\sum_i^n (x_i - \overline{x})(y_i - \overline{y})}{\sqrt{\sum_i^n (x_i - \overline{x})^2} \sqrt{\sum_i^n (y_i - \overline{y})^2}}, \tag{2}$$

where $x_i$, $y_i$ are a pair of variables for the $i^{th}$ set of data and $\overline{x}$, $\overline{y}$ are the means of variables $x$, $y$ from all observations ($n$). The full set of data was used to perform the correlation analysis, and the inter-relationships between the variables is presented in Fig. 1-b. Directed arrows were drawn from each variable to all of the variables that showed a correlation coefficient greater than 0.35.

The probability of fracture event occurrence in extreme metocean conditions (*i.e.*, conditions where the values of the

model variables were extremely high) was investigated using a full set of model criteria with the high state ($> x^*$) of each variable. *States* here refer to the variable intervals defined based on a threshold. The selected criteria ($x^*$) for the extreme condition of each variable was identified by varying each criterion over the range of each variable from the fracture subset to maximize the fracture event probability. The distribution of conditional posterior probability was calculated through Bayes' Theorem, given by Stuart and Ord (1994):

$$P(X|Y) = \frac{P(X) \times P(Y|X)}{P(Y)},$$ (3)

where $P(X)$ is the prior probability of fracture event occurrence, $P_{frac}$; $P(Y|X)$ is the likelihood of a specific criteria set occurrence during fracture events, $P(V_w, T_a, V_c, T_w, E_w, T_{a\_avg}, T_{w\_avg}|Frac)$; and $P(Y)$ is the evidence of the criteria set occurrence for all observations, $P(V_w, T_a, V_c, T_w, E_w, T_{a\_avg}, T_{w\_avg})$. The probabilities in Eq. (3) should be recalculated when new evidence become available, a process that reduces the dependence of the posterior probability on the original estimated prior probability (Eleye-Datubo et al., 2006). Given the large size of the CI2D3 database, the value of $P(X)$ was estimated as the frequency of fracture events (*i.e.*, the number of fracture events divided by the total number of observations) before any criteria set based on metocean conditions was considered. The values of $P(Y|X)$ and $P(Y)$ were determined using the relative frequency of the set of states in fracture events and all observations, respectively. The relative frequency is given by Bonafede and Giudici (2007):

$$P(S_i) = \frac{n_i}{n},$$ (4)

where $S_i$ represents a set of the variables' states, $n_i$ represents the count of the observed set of the states in fracture events (or all observations), and $n$ represents the total number of fracture events (or all observations) in the dataset.

To calculate the probability of fracture events in different metocean conditions, the ranges of atmospheric and oceanic variables at the time of fracture events were first divided into two states using the variable distribution medians from the fracture subset (Table 1). The conditional fracture probabilities were then estimated using a similar Bayesian approach (Eq. (3)) through analyzing concurrent atmospheric and oceanic conditions at the time of fracture events that were extracted from the entire record for all ice island observations. Due to the limited number of fracture events (328), the number of state combinations in the presented model needed to be reduced to avoid model saturation and increase the model reliability. Among the atmospheric and oceanic variables analyzed in this study, current speed played an insignificant role in the fracture events of the ice islands, so it was not considered for further analysis.

**Table 1.** Various states of atmospheric and oceanic variables for the Bayesian fracture model

| Variables | Median [1] | Unit | State 1 | State 2 |
|---|---|---|---|---|
| Wind Speed ($V_w$) | 2.8 | $m\ s^{-1}$ | $\leq 2.8$ | $> 2.8$ |
| Air Temperature ($T_a$) | $-2.1$ | °C | $\leq -2.1$ | $> -2.1$ |
| Water Temperature ($T_w$) | $-0.3$ | °C | $\leq -0.3$ | $> -0.3$ |

| | | | | |
|---|---|---|---|---|
| Wave Energy Index ($E_w$) | 5.1 | $m^2\,s$ | $\leq 5.1$ | $> 5.1$ |
| Lifetime Mean Air Temperature ($T_{a\_avg}$) | $-3.5$ | ℃ | $\leq -3.5$ | $> -3.5$ |
| Lifetime Mean Water Temperature ($T_{w\_avg}$) | $-0.7$ | ℃ | $\leq -0.7$ | $> -0.7$ |

[1] Median values in the distributions of the given variables from the fracture subset

The developed probabilistic model was validated using a resampling approach based on the Pareto principle (Macek, 2008), which suggests 80% of the data be used for model training and development, and 20% be reserved for testing the developed model (Suthaharan, 2016). To reduce the effect of variation in the subset selection and have a more robust evaluation of the developed model, a k-fold cross-validation approach (Ozdemir, 2016) was used (k=5). So, input variables associated with the fracture and non-fracture data were randomly partitioned into five disjoint subsets of approximately equal size, where each time one of these subsets served for model testing, and the rest were used to train the model. This corresponded to the selection of training subsets with approximately 14204 data points (262 fracture events and 13942 non-fracture events) and test subsets with approximately 3551 data points (66 fracture events and 3485 non-fracture events). The conditional fracture probabilities of ice islands for the given criteria sets were calculated using the atmospheric and oceanic conditions for each test subset and then cross-validated against the predicted values associated with its corresponding training set.

## 3 Results and discussion

### 3.1 Preliminary analysis of ice island fracture events

The descendants of ice islands resulting from the calving events of Petermann Glacier in 2008, 2010, 2011, and 2012 generally drifted in a southward direction toward the Labrador Sea (Fig. 2-a). These ice islands experienced 328 fracture events (Fig. 2-b), which resulted in 845 ice islands. The 2010 event calved the largest ice island (Table 2), which generated 637 ice islands through its fractures (242 times), some of which drifted as far as offshore Newfoundland (Fig. 2-a). The second largest calving event happened in 2012 which generated 169 ice islands through 73 fracture events, but the resulting ice islands were only recorded as far as offshore Iqaluit, given that the monitoring period in the CI2D3 database ended in December 2013. The other two calving events (2008 and 2011) generated 29 and 10 ice islands, which resulted from nine and four calving events, respectively. The size distribution of ice islands showed that large ice islands (>10 km$^2$) drifted longer before undergoing a fracture event and split into greater numbers of pieces per fracture event. Examples of this are two large ice islands (~137 km$^2$ and 60 km$^2$) originating from the 2010 calving event, which generated nine distinct pieces upon fracturing. However, around 70% of all fracture events generated only two children ice islands. A more detailed drift and deterioration analysis of the Petermann ice islands is presented in Zeinali-Torbati et al. (2019).

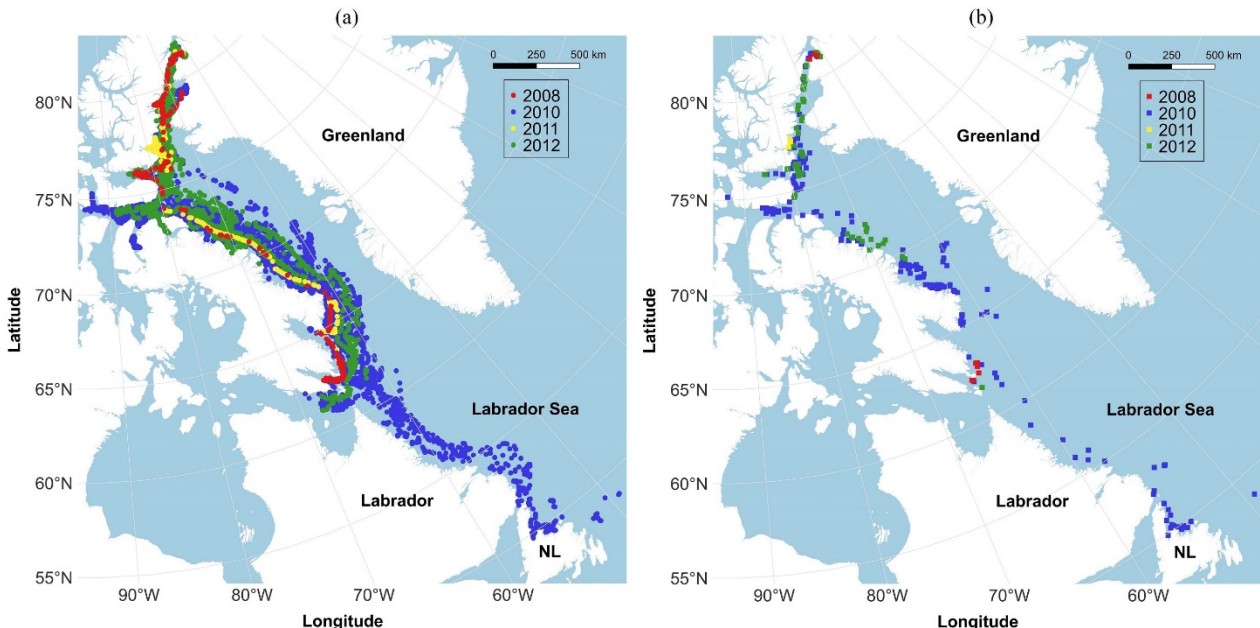

**Figure 2.** The drift trajectories (a) and the locations of fracture events (b) for the ice islands originating from the calving events of the Petermann Glacier in 2008 (red), 2010 (blue), 2011 (yellow), and 2012 (green)

**Table 2.** Description of the ice islands originating from the massive calving events of Petermann Glacier in 2008, 2010, 2011, and 2012

| Glacier calving year | Main calving date | Surface area (km²) | Number of digitized polygons | Number of fracture events |
|---|---|---|---|---|
| **2008** | 10 July | 36.4 | 332 | 9 |
| **2010** | 5 August | 302.4 | 9658 | 242 |
| **2011** | 16 August 21 September | 4.3 | 502 | 4 |
| **2012** | 17 July | 144.6 | 7263 | 73 |

### 3.2 Distributions of atmospheric and oceanic variables

The atmospheric, oceanic, and lifetime mean variables shown in Fig. 1 were examined at the time of ice island fracture events, and then compared with the metocean conditions for all ice island observations using the methodology described earlier. Figure 3 through Fig. 5 show the summary statistics and histogram plots of the relative frequency of regional metocean variables surrounding the Petermann ice islands from all observations (blue), as well as from the fracture events (red).

Figure 3-a shows that the mean water temperature surrounding all Petermann ice islands was negative (-0.8 ℃)

indicating that they mainly drifted within cold waters; however, the water temperature values reached up to 10.4 ℃. The statistics for lifetime mean water temperature from all observations (Fig. 3-c) were almost the same as water temperature: a negative mean value of -0.8 ℃ and a range of -1.8 ℃ to 10.4 ℃. Comparing the distribution of water temperatures in Fig. 3-a,b reveals that fracture events happened at higher water temperatures; while only 20% of ice islands from the entire dataset were surrounded by water temperatures above 0 ℃, 42% of fractured ice islands were subjected to positive water temperatures. In a similar way, it was revealed from long-term water temperature distributions that only 13% of the ice islands from all observations drifted in positive lifetime mean water temperatures (Fig. 3-c). This, however, corresponded to about 34% of the ice islands in the fracture subset (Fig. 3-d). The summary statistics presented in Fig. 3 reveal that water temperature and lifetime mean water temperature played significant roles in the fracture events of Petermann ice islands; compared to the temperature records for all ice island observations, the ice islands at the time of fracture events experienced, on average, 1.2 ℃ and 0.8 ℃ greater values of water temperature and lifetime mean water temperature, respectively. This indicates the important contribution of warm waters to faster deterioration of glacial ice features (as stated by Kubat et al., 2007), likely due to higher internal stress caused by the increased heat transfer from water to the ice feature. The significant contribution of water temperature to fracturing process was corroborated by Bouhier et al. (2018), where a significant correlation between iceberg relative volume loss and sea surface temperature was found. Warm surface waters also plays an important role in the initiation of fractures on large tabular Antarctic icebergs (England et al., 2020) through edge-wasting (*c.f.*, Scambos et al., 2005).

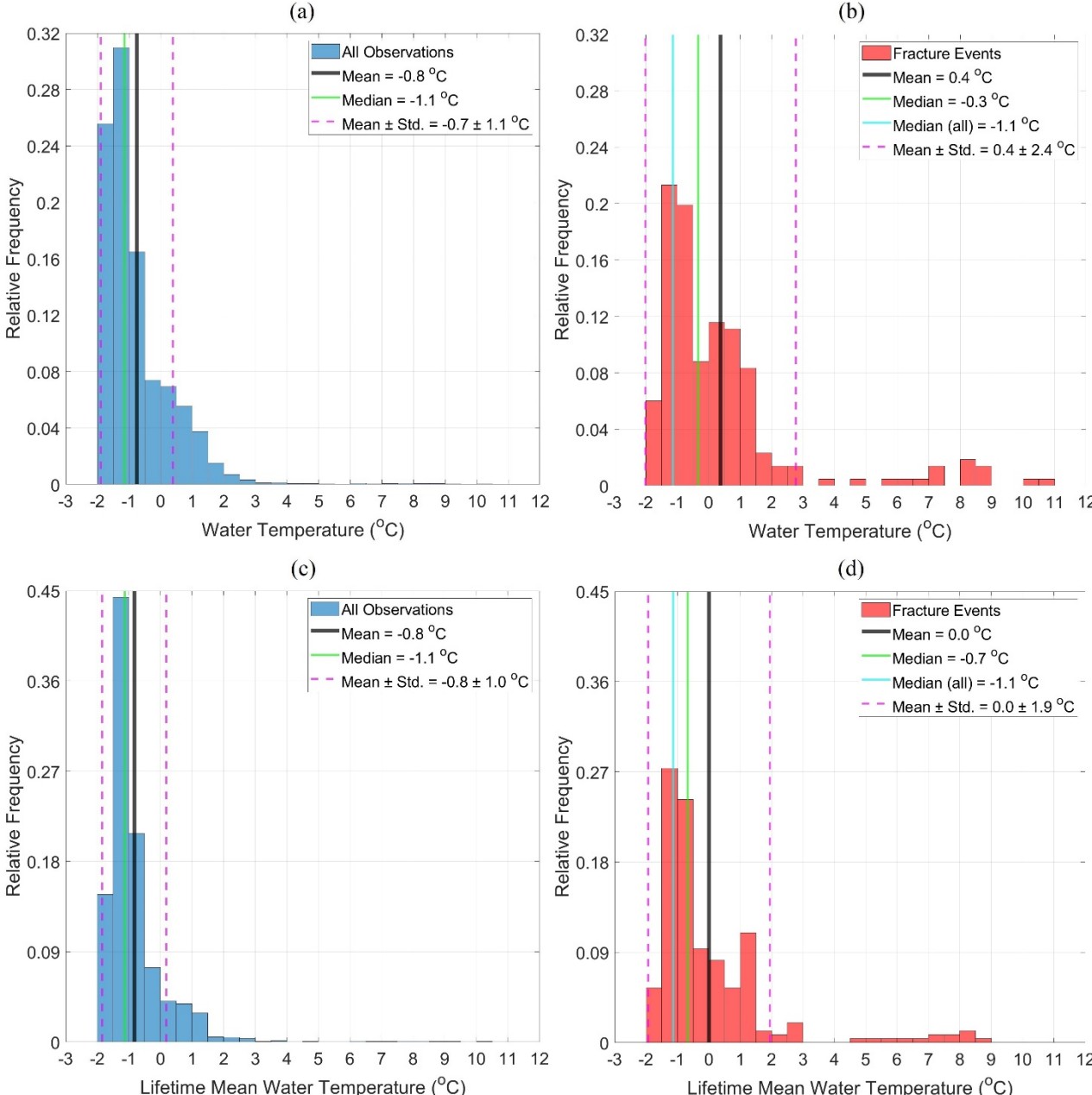

**Figure 3.** Relative frequency histogram plots of water temperature (a; n=16784, b; n=298) and lifetime mean water temperature (c; n=13537, d; n=256) surrounding Petermann ice islands for all observations (a,c), and for the fracture events (b,d)

Figure 4-a shows that the air temperatures to which the ice islands were subjected ranged from -35.6 ℃ to 17.9 ℃ for all observations, but the mean air temperature of -9.4 ℃ reveals that the ice islands drifted a significant amount of time in cold air temperatures. Similarly, the Petermann ice islands were mainly subjected to negative lifetime mean air temperatures with

an average of -11.6 ℃ ranging from -33.7 ℃ to 17.9 ℃ (Fig. 4-c). Investigation of air temperature and its long-term effect at the time of fracture events (Fig. 4-b,d) revealed that, on average, the ice islands were subjected to much higher values of these variables. The average air temperature at the time of fracture events was -5.2 ℃, which was 4.2 ℃ higher than the mean air temperature surrounding all ice islands. In a similar way, the ice islands from the fracture subset were, on average, subjected

to 5.7 ℃ higher lifetime mean air temperature than the ice islands from all observations. The air temperature distributions (Fig. 4-a,b) show that while the air temperatures associated with all ice island observations were most frequent around 0 ℃, the values associated with the fracture events were most frequent between 0 ℃ and 4 ℃. This, along with a 4.2 ℃ higher mean air temperature value from the fracture subset indicate that higher air temperature values are likely linked with the occurrence of fracture events. The analysis of lifetime mean air temperature distributions over the lifespan of each ice island (Fig. 4-c,d)

revealed that while only 14% of all ice island observations experienced lifetime mean air temperatures greater than 0 ℃, about 35% of the ice islands from the fracture subset were subjected to positive lifetime mean air temperatures. This indicates that the long exposure of ice islands to relatively warm air temperatures was likely an important factor in the occurrence of the fracture events.

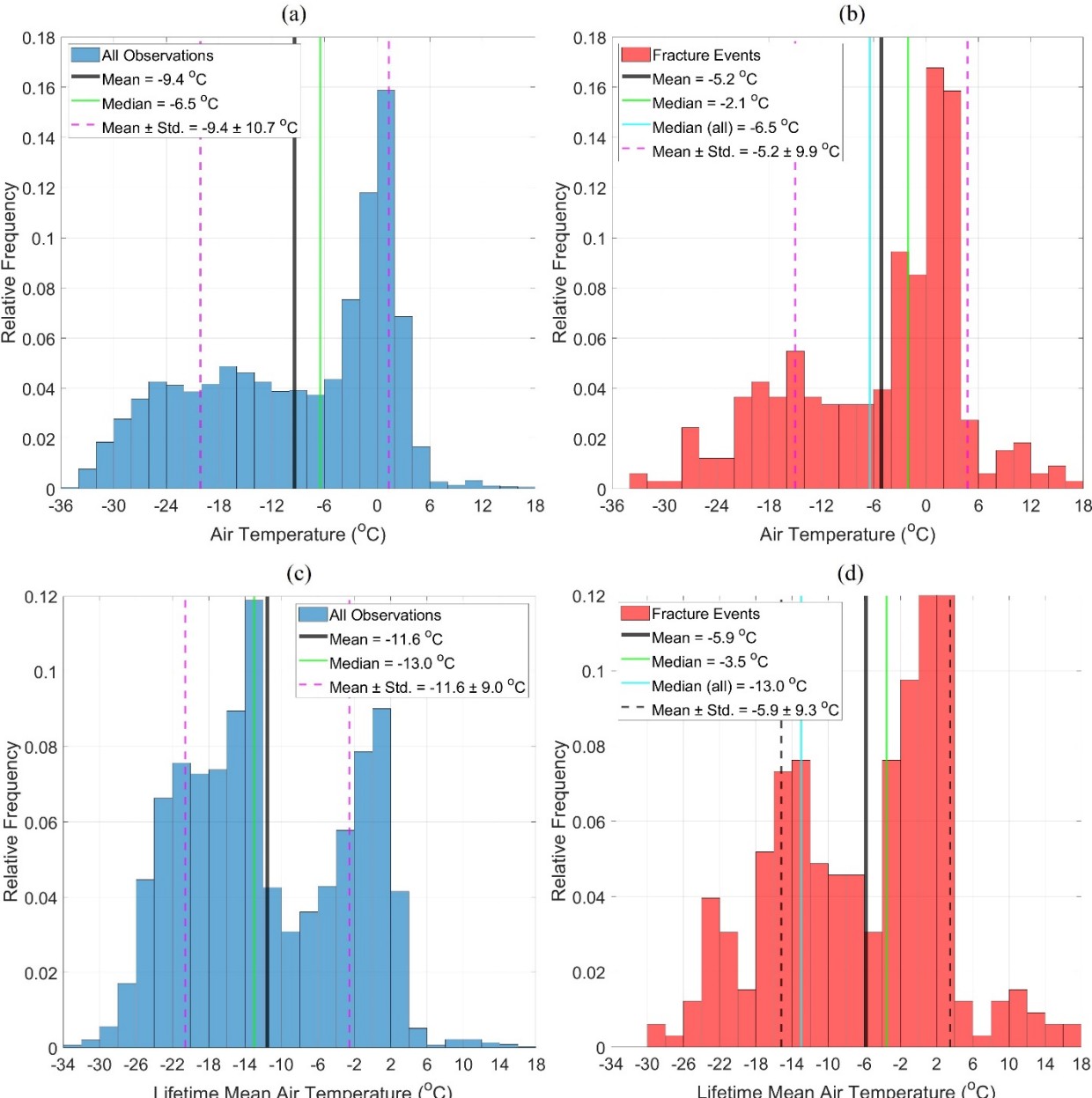

**Figure 4.** Relative frequency histogram plots of air temperature (a; n=17735, b; n=328) and lifetime mean air temperature (c; n=17755, d; n=328) surrounding Petermann ice islands for all observations (a,c), and for the fracture events (b,d)

The results associated with all observations of waves (Fig. 5-a) show that while the wave energy index values varied from $0.1\ m^2\ s$ to $62.1\ m^2\ s$, the ice islands were mainly subjected to relatively low wave energy index with an average value of $5.3\ m^2\ s$. Similarly, the regional wind and current speeds surrounding the ice islands from all observations were often relatively low, with a mean value of $2.9\ m\ s^{-1}$ and $0.08\ m\ s^{-1}$, respectively (Fig. 5-c,e). The summary statistics of the records

from the fracture events (Fig. 5-b,d,f) revealed that the regional mean wave energy index, wind speed, and current speed around the ice islands at the time of fracture events were statistically higher (by 38%, 17%, and 38%, respectively) when compared to the associated values for all ice islands observations. The analysis of wave energy index distributions around the Petermann ice islands presented in Fig. 5-a,b show that only 30% of Petermann ice islands encountered wave energy index values greater

than 6 $m^2\ s$. The fracture events, however, occurred at greater wave energy index values, where 46% of ice island observations experienced wave energy index values greater than 6 $m^2\ s$. The analysis also shows that while relatively high values of wave energy index, coupled with other metocean variables, most likely contribute to the occurrence of ice island fracture events, this variable by itself does not lead to a high fracture probability of Petermann ice islands. This is consistent with the results of the iceberg deterioration study by Bouhier et al. (2018) where the authors found no significant link between the relative

volume loss of two large Antarctic icebergs and the wave-related variables. Investigation of wind speeds over the ice islands (Fig. 5-c,d) showed that the ice islands from all observations were subjected most frequently to weak winds (~1-3 $m\ s^{-1}$). Similarly, at the time of fracture events, the ice islands were most frequently subjected to weak winds (~2-4 $m\ s^{-1}$). The fact that there is little difference in these distributions suggests that wind speed by itself was not a significant variable in the fracture of ice islands. The comparison of ocean current speed records around the studied ice islands (Fig. 5-e,f) revealed that the

current speed values from the fracture subset were statistically greater than the full observational records. However, similar to all records, the fractured ice islands were mainly subjected to weak regional current speeds, which suggests minor contribution of current speed to the fracture event occurrence.

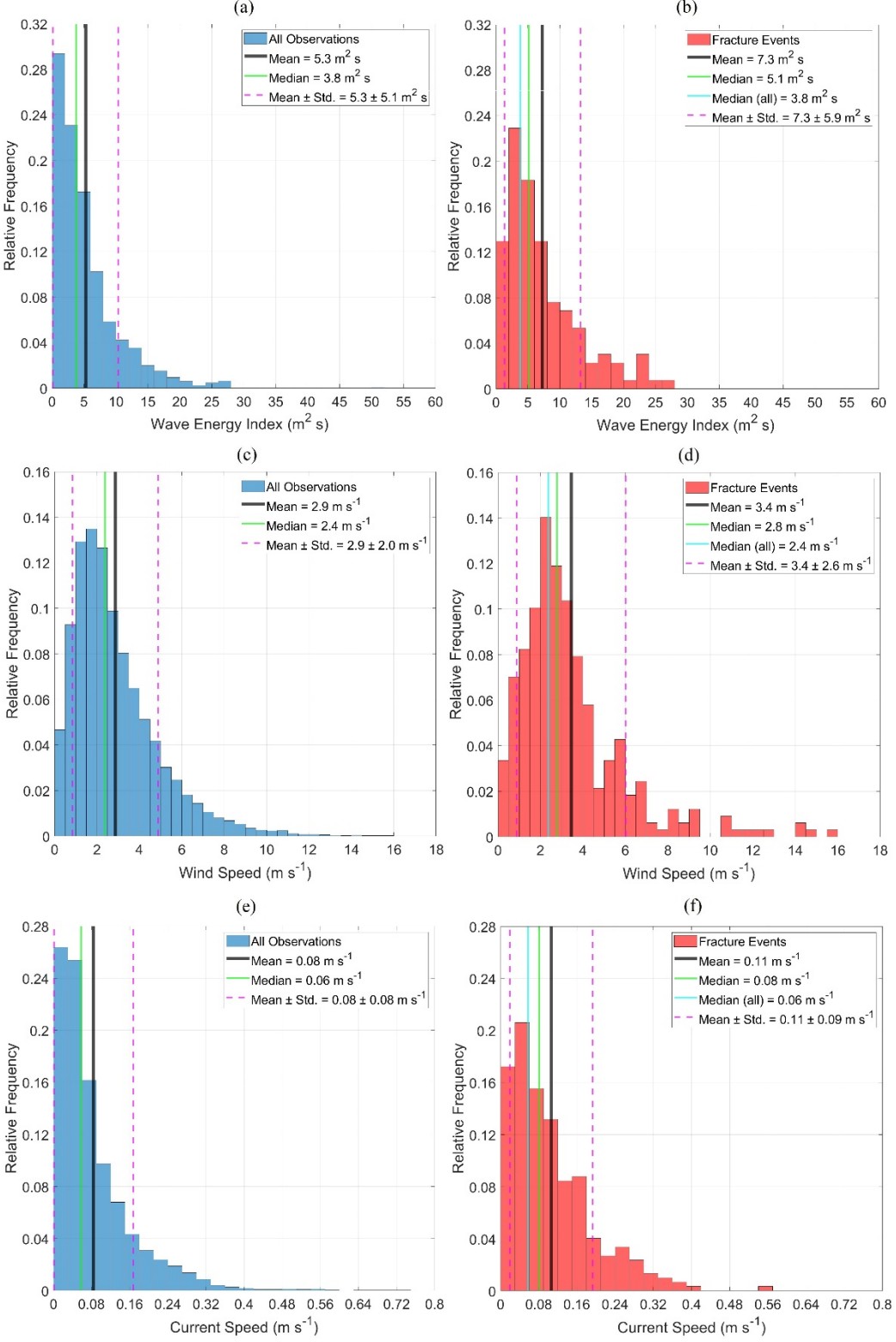

**Figure 5.** Relative frequency histogram plots of wave energy index (a; n=3985, b; n=131), wind speed (c; n=17735, d; n=328), and current speed (e; n=16791, f; n=296) surrounding Petermann ice islands for all observations (a,c,e), and for the fracture events (b,d,f)

The enhancement of fracture events under the conditions where the ice islands experienced higher values of metocean variables was investigated through ratios of the relative frequency for fracture events and all observations over the range of variables presented in Figs. 3-5. These results are presented in Appendix A (Figs. A1-A3), where values close to one imply that fracture events are as likely to occur as the frequency of observations. Values large compared to one indicate that fracture events are more likely to occur than the frequency of observations. Values less than one imply that fracture events are less likely to occur relative to the frequency of observations. The results in Figs. A1-A3 reveal that the ratio of the relative frequency for fracture events and all observations generally increases with the values of metocean variables, which clearly indicate a tendency for fracture events to occur under more extreme conditions.

The pairwise correlation between the metocean variables revealed that water temperature was positively correlated with the air temperature, a finding that was stated in a number of other studies (*e.g.*, Morrill et al., 2001; Erickson and Stefan, 2000). Table 3 also revealed a correlation between the wind speed and wave energy index, which was expected as the wave energy index is dependent on the significant wave height (Eq. (1)). The positive correlation between wind speed and wave height was also stated in Fu et al. (2016). Other inter-relationships between the variables include the correlations between the daily-average air/water temperatures and the lifetime mean air/water temperatures. These correlations are expected given that the lifetime mean variables were defined as the time-average of daily-average variables over the life of each ice island.

**Table 3.** Pearson product-moment correlation coefficients of the metocean variables in the developed fracture model. The variables included water temperature ($T_w$), wind speed ($V_w$), air temperature ($T_a$), current speed ($V_c$), wave energy index ($E_w$), lifetime mean air temperature ($T_{a\_avg}$), and lifetime mean water temperature ($T_{w\_avg}$)

| Variable | $V_w$ | $T_a$ | $V_c$ | $T_w$ | $E_w$ | $T_{a\_avg}$ | $T_{w\_avg}$ |
|---|---|---|---|---|---|---|---|
| $V_w$ | 1 | | | | | | |
| $T_a$ | 0.132 | 1 | | | | | |
| $V_c$ | 0.097 | 0.079 | 1 | | | | |
| $T_w$ | 0.236 | 0.547 | 0.076 | 1 | | | |
| $E_w$ | 0.403 | 0.046 | 0.066 | 0.232 | 1 | | |
| $T_{a\_avg}$ | 0.216 | 0.650 | 0.145 | 0.581 | 0.097 | 1 | |
| $T_{w\_avg}$ | 0.265 | 0.366 | 0.153 | 0.847 | 0.297 | 0.647 | 1 |

The associated p-values for all correlations show significance at the level of 0.00005 (p-value<0.00005).

It should be noted here that correlation does not imply causation, but there is clearly an element of causation within the correlations noted above. Given the positive correlation between the temperature variables ($T_a, T_w, T_{a\_ave}, T_{w\_ave}$), it is

expected that ice islands exposed to warm water temperatures also experience warm air temperatures and lifetime mean air and water temperatures, conditions that contribute to ice melting. The resulted ablation could lead to ice island gravitational change and trigger fracture events. Similarly, the positive correlation between $V_w$ and $E_w$ implies a high chance for simultaneously high states of wind speed and wave energy index. These variables are linked to the external forces acting on ice islands. While, strong winds increase the associated wind drag forces, large values of wave energy index result in higher forces from waves. These forces together contribute to the accumulation of stress in the ice island, which could ultimately exceed the local threshold of fracture energy and result in fracture events. Wave actions also play a role via the "footloose mechanism" (Wagner et al., 2014), which could result in the instability of the ice geometry and potentially lead to ice island fracture events.

### 3.3 Metocean conditional criteria sets and fracture event frequency

To examine the probability of fracture event occurrence in very high states of metocean conditions, a full set of model criteria associated with the atmospheric and oceanic conditions at the time of fracture events was obtained and presented as criteria set i=6 (Table 4), using the methodology described earlier. The same approach was employed to investigate the influence of simplifying the fracture model using fewer number of variables, and the results were presented by criteria sets i=1-5 in Table 4.

**Table 4.** Fracture model conditional criteria sets and the associated conditional probability ($P_{frac}$) for extreme conditions. The variables included water temperature ($T_w$), wind speed ($V_w$), air temperature ($T_a$), current speed ($V_c$), wave energy index ($E_w$), lifetime mean air temperature ($T_{a\_avg}$), and lifetime mean water temperature ($T_{w\_avg}$)

| Criteria Set i | $T_w$ (°C) | $V_w$ ($m\ s^{-1}$) | $T_a$ (°C) | $V_c$ ($m\ s^{-1}$) | $E_w$ ($m^2\ s$) | $T_{a\_avg}$ (°C) | $T_{w\_avg}$ (°C) | $P_{frac}$ (%) |
|---|---|---|---|---|---|---|---|---|
| 1 | >4 | | | | | | | 16 |
| 2 | >4 | >6 | | | | | | 28 |
| 3 | >4 | >6 | >7 | | | | | 45 |
| 4 | >4 | >6 | >7 | >0.1 | | | | 60 |
| 5 | >4 | >6 | >7 | >0.1 | >5 | | | 75 |
| 6 | >4 | >6 | >7 | >0.1 | >5 | >0 | >0 | 75 |

Table 4 has a predictive capability for the occurrence of fracture events for the Petermann ice islands under different extreme atmospheric and oceanic conditional criteria sets. For example, if the conditions associated with the criteria set i=2 hold (*i.e.*, water temperatures greater than 4 °C and wind speeds greater than 6 $m\ s^{-1}$), there is a 28% chance that these conditions lead the ice islands to fracture. To elaborate, criteria set i=2 accounts for only water temperature and wind speed, where there are five events from the fracture subset (328 events) and 18 events from all observations (17755 events) that meet

the given criteria. Therefore, a conditional fracture probability of 0.28 was obtained for this criteria set, as follows:

$$P(Frac|T_w > 4\,°C, V_w > 6\ m\ s^{-1}) = \frac{P_{frac} \times P(T_w > 4\,°C, V_w > 6\ m\ s^{-1}|Frac)}{P(T_w > 4\,°C, V_w > 6\ m\ s^{-1})} = \frac{\frac{328}{17755} \times \frac{5}{328}}{\frac{18}{17755}} \approx 0.28 , \qquad (5)$$

Some features of the criteria sets presented in Table 4 and their associated fracture event probability are noteworthy. An important implication of the results in Table 4 is the predominant link between the daily-average variables (*i.e.*, $T_w$, $V_w$, $T_a$, $V_c$, and $E_w$) with the fracture event occurrence. When only the criteria for the air and water temperatures, wind and current speeds, and wave energy index are considered without accounting for the lifetime mean variables (criteria set i=5), 75% of all events meeting these criteria occurred for the ice islands from the fracture subset. However, when only the high states of one or two variables are considered, the fracture probability drops to less than 30%, which indicates the strong effect of concurrent atmospheric and oceanic conditions on the occurrence of ice island fracture events. Also, the criteria set i=6 in Table 4 reveals that, the addition of the lifetime mean variables did not increase the fracture probability above 75%. This is due to the fact that the criteria set i=5 narrowed down the atmospheric and oceanic conditions to a condition that already meets the criteria added in criteria set i=6, implying that the conditions presented in criteria set i=5 are enough to predict a fracture event probability up to 75%. It is worth noting that the criteria sets represented in Table 4 implicitly account for the inter-relationship between the variables. For example, the criteria set i=3, which considers high air and water temperatures, accounts for the correlation between air and water temperatures (Table 3), indicating that there is a high chance for the coincident occurrence of high air and water temperatures. It is also important to note that Table 4 only shows the criteria sets associated with the fracture events of ice islands originating from the calving events of the Petermann Glacier in 2008, 2010, 2011, and 2012. If more data become available from other fracture events of the Petermann ice islands, where the fracture events occur under different combinational criteria sets of atmospheric and oceanic conditions, then the variable ranges, the number of criteria sets, and the fracture probabilities presented in Table 4 would need to be updated. Under such conditions, the model variables themselves could also be modified if additional variables (*e.g.*, sea ice concentration) were deemed to be important.

The atmospheric and oceanic conditional criteria sets presented in Table 4 only show some specific criteria sets for extreme conditions and clearly do not represent all possible combinations of the conditions. These criteria sets were selected by the progressive addition of one or two conditions to the previous criteria set, so that the associated conditional fracture probability would increase. However, to account for all possible conditions of the variables, the ranges for the atmospheric and oceanic variables were divided into two states using the methodology described earlier in Table 1. The elimination of current speed variable (as explained in the Sect. 2.3) reduced the number of state combinations from 128 to 64, which allowed for a greater number of occurrences for each combination of the states, with bin edges previously described in Table 1. Through an iterative process, the conditional fracture probability for each combination of the states was then calculated using Bayes' Theorem (Eq. (3)). Due to the large number of elements (64 values) in the conditional probability table, it is not possible to illustrate all combinations of the variable states (Table 1) and their associated probability. So, the model description is limited to its qualitative part.

The results obtained from the presented Bayesian approach provide a framework to probabilistically forecast the future fracture events of ice islands originally calved from the Petermann Glacier. This probabilistic model can provide supplementary information to the available deterministic ice dynamic prediction models by quantifying the probability of ice island fracture event occurrence under various sets of concurrent atmospheric and oceanic conditions. To use this model, one needs to first identify the spatial and temporal coordinates of a given ice island and then extract or forecast the six daily-average and lifetime mean variables discussed earlier. Then, the obtained set of conditions is identified in the developed conditional probability table to quantify the associated probability of fracture event occurrence under the given set of atmospheric and oceanic conditions.

## 3.4 Case study

To better describe the utility of the developed fracture model, a case study was conducted on a descendant of ice islands resulting from the calving event of Petermann Glacier in 2010. This ice island was selected for a case study due to its drift characteristics. The ice island drifted for a long period (26 November 2010 – 5 July 2011; >7 months) and experienced a significant change in latitude (>15º). The spatial and temporal data for consecutive observations of this ice island were identified using the CI2D3 database. The atmospheric and oceanic variables were extracted from reanalysis databases and then interpolated to the positions and times of the given observations. The lifetime mean variables were also estimated using the methodology explained earlier, and the variables were all used as input to the presented fracture model to study the conditional fracture probabilities of the ice island over its drift path (Fig. 6-a). To investigate the effect of different atmospheric and oceanic conditions on the fracture events of the same ice island, metocean data from 2017-2018 were extracted and interpolated to the same positions as the case study ice island. Here it was assumed that the same drift trajectory applies, however, this would not likely be the case given that iceberg drift models are largely governed by the real-time metocean conditions (Lichey and Hellmer, 2001; Kubat et al., 2005; Eik, 2009; Keghouche et al., 2009; Rackow et al., 2017; Zeinali-Torbati et al., 2020). This would be, however, an area for future work, where a drift model needs to be integrated into the proposed fracture model to have a reliable estimation of ice island positional data. The interpolated metocean data for the 2017-2018 case were then used in the presented fracture model to investigate how the fracture probability map would change under the influence of different atmospheric and oceanic conditions (Fig. 6-b). The fracture probability map for the 2010-2011 case (Fig. 6-a) reveals that the ice island drifted some time (~14 days) in the medium-high fracture probability zone (shown by the orange and red colors around Labrador coast) before breaking up into two pieces. This, however, was not the case for the 2017-2018 ice island during the same time period (Fig. 6-b) as this ice island only spent some time (~14 days) in the small-medium fracture probability zone (shown by the green and orange colors) towards the end of its hypothetical drift off Labrador coast, which could thus likely have been longer than the 2010-2011 drift. The hypothetical 2017-2018 ice island, in fact, never experienced the metocean conditions that lead to a probability of fracture <4%. This is likely due to the fact that the 2017-2018 ice island was generally exposed to lower water temperature (~0.3 ℃ on average) over its drift. This result is expected given that water temperature was identified as the most important contributor to the fracture events of Petermann ice islands analyzed in this

study.

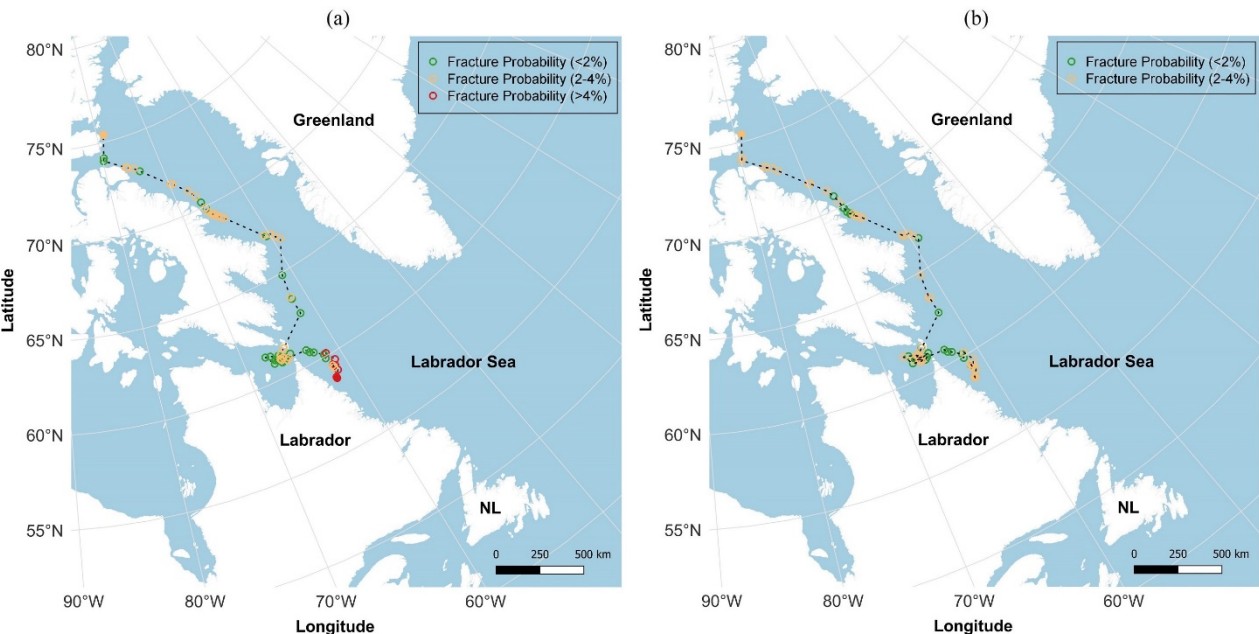

**Figure 6.** The fracture probability map for a descendant of the Petermann ice island from the 2010 calving event (a), compared against the projected fracture probabilities for the same ice island in 2017-2018 (b). The filled dots show the positions of the
485 ice island at the time it was born and the time it fractured

While a Bayesian network has never been employed for forecasting ice island fracture events, the probabilistic model presented in this paper was developed based on the methodology used in Turnbull et al. (2019) and Fu et al. (2016), where the Bayesian approach was used to predict vessel besetting events in pack ice. The study by Fu et al. (2016) used a Bayesian network to investigate the inter-relationship between nine variables (*i.e.*, ship speed, engine power, wind speed, air
temperature, low visibility, sea temperature, ice concentration, ice thickness, and wave height), as well as their influence on the probability of a ship getting stuck in ice while navigating through the Northern Sea Route. Using a similar Bayesian approach, Turnbull et al. (2019) studied two pack ice besetting events of the Umiak I and developed a probabilistic forecast model for future besetting events experienced by Umiak I under the influence of nine ice and metocean variables (*i.e.*, ice concentration, ice thickness, floe size, minimum coast distance, wind-coast direction, wind speed, current-coast direction,
current speed, and wind divergence). While this approach has not been used in a past iceberg fracture model, there are some deterioration models such as the one by Kubat et al. (2007) that account for the influence of metocean variables (*e.g.*, wind speed, current speed, water temperature, wave height, and wave period) on the calving events of the overhanging slabs resulting from the repeated action of waves. Kubat et al. (2007) revealed that wave height and water temperature dominate effects on iceberg deterioration through melt and small-scale wave-induced calving events. However, the probabilistic model presented
here accounts for the large-scale fracture events in ice islands under the influence of various metocean conditions that govern the occurrence of these events. To date, there has been limited research (*e.g.*, Bouhier et al., 2018) investigating the atmospheric

and oceanic conditions that lead to the highest probability of large-scale iceberg fracture event occurrence. While the bulk volume loss model by Bouhier et al. (2018) can provide a representation of iceberg relative volume loss variation with sea surface temperature and iceberg velocity, it is not able to estimate iceberg fracture probability under the influence of environmental variables presented here. Therefore, it is difficult to provide a validation scheme that can appropriately compare the results of the presented Bayesian fracture model with an existing physical ice island fracture model. Hence, the presented model was validated using a well-known scheme in probabilistic data analysis studies, which was described in Sect. 2.3.

## 4 Probabilistic model validation

The developed fracture model was validated using k-fold cross-validation approach described in Sect. 2.3, and the results were presented in Table 5. The model validation analysis for the selected criteria sets in Table 5 reveal that the mean fracture probabilities estimated from the test subsets are in agreement with the mean estimations from the training subsets as the ranges for the fracture probability values overlap. Investigation of the errors between the pairwise (test vs. training) fracture probability values show that the test sets selected using the 5-fold cross-validation approach are able to provide estimations that are, on average, within 13-39% of the values forecasted using the training sets. It was also revealed from the standard deviation values that the presented model is more reliable when a fewer number of variables or state combinations are considered (*e.g.*, criteria set i=1). This is because, as the number of variables or state combinations increases, there are fewer fracture events and the ranges of atmospheric and oceanic conditions become more constrained, so the number of events meeting these given criteria decreases. With fewer observations, there will be more variability and consequently a higher error in the predicted fracture probability values. If more fracture data in each criteria set become available, the model will become more robust and the error in fracture probability estimations is expected to reduce.

**Table 5.** Model validation for some of the criteria sets in the model (*e.g.,* criteria set i=5 shows one of the 64 state combinations of the six variables; criteria set i=4 represents one of the 16 state combinations of the four variables). The variables included water temperature ($T_w$), wind speed ($V_w$), air temperature ($T_a$), wave energy index ($E_w$), lifetime mean air temperature ($T_{a\_avg}$), and lifetime mean water temperature ($T_{w\_avg}$). Model error is derived through statistical comparison of fracture probability estimations from training sets and test sets, obtained using 5-fold cross-validation method

| Criteria Set i | $T_w$ (°C) | $V_w$ ($m\,s^{-1}$) | $T_a$ (°C) | $E_w$ ($m^2\,s$) | $T_{a\_avg}$ (°C) | $T_{w\_avg}$ (°C) | $P_{frac\_training}$[1] (%) Mean | Std.[3] | $P_{frac\_test}$[1] (%) Mean | Std.[3] | *Pairwise* % *Error*[2] Test vs. Training | $P_{frac\_all}$[1] (%) |
|---|---|---|---|---|---|---|---|---|---|---|---|---|
| 1 | >-0.3 | | | | | | 3.3 | 0.1 | 3.4 | 0.4 | 13 | 3.3 |
| 2 | >-0.3 | >2.8 | | | | | 3.4 | 0.2 | 3.5 | 0.9 | 20 | 3.5 |
| 3 | >-0.3 | >2.8 | >-2.1 | | | | 3.8 | 0.3 | 3.8 | 1.2 | 24 | 3.8 |
| 4 | >-0.3 | >2.8 | >-2.1 | >5.1 | | | 6.1 | 0.6 | 6.0 | 2.4 | 33 | 6.2 |
| 5 | >-0.3 | >2.8 | >-2.1 | >5.1 | >-3.5 | >-0.7 | 6.5 | 0.9 | 6.6 | 3.4 | 39 | 6.7 |

[1] $P_{frac\_training}$, $P_{frac\_test}$, and $P_{frac\_all}$ represent the fracture probability estimations from the training subsets, test subsets,

and all data points, respectively

[2] Relative error between the fracture probability estimations from the training and test subsets

[3] Standard deviation

The 5-fold cross-validation analysis presented in Table 5 only shows some of the possible combinations of variable states that were defined based on the median values of the model variables from the fracture subset (presented in Table 1). This corresponds to 1/2 one-variable combinations (i=1), 1/4 two-variable combinations (i=2), 1/8 three-variable combinations (i=3), 1/16 four-variable combinations (i=4), and 1/64 six-variable combinations (i=6). However, the model skill was also analyzed for the remaining combinations, and it was revealed that the model does not perform well under implausible

combinations of the atmospheric and oceanic conditions, which are not likely to be encountered and do not hinder the model most of the time. As an example, the condition where $T_w > -0.3\,°C$, $V_w > 2.8\,m\,s^{-1}$, $T_a \leq -2.1\,°C$ ), $E_w > 5.1\,m^2\,s$, $T_{a\_avg} > -3.5\,°C$, and $T_{w\_avg} \leq -0.7\,°C$ was a very unlikely combination that only occurred once among all ice island observations, and no fracture event occurred under such conditions. Based on the extracted/interpolated metocean data for the full model with all six variables, 36 combinations (out of 64) never occurred, so the fracture probabilities under such conditions

are unknown. However, the remaining 28 combinations that were met revealed a larger error (~100-200%) between the probability estimations from the training and test subsets, when the associated combination was unlikely to occur (<1%). For instance, due to the very few data points existing for these improbable combinations, there were some cases that were not observed in the test subsets but were observed only a few times in the training subsets, which inflated an error of 100%. However, our model showed higher reliability under plausible combination of metocean conditions, such as the criteria sets

i=1-5 in Table 5 (13-39% error), or when fewer number of variables were used in the model that generated much less error between the probability estimations from the test and training subsets (*e.g.*, 11-97% for 1-4 variables).

**5 Conclusions and future work**

This study presented a probabilistic forecast model for the fracture events of ice islands through the analysis of the relative influences of atmospheric and oceanic forces. The recurrent deterioration of the ice islands originating from four recent calving

events of Petermann Glacier were studied using the data in the CI2D3 database to probabilistically investigate the conditions that lead to fracture event occurrence of the ice islands. It was revealed in Fig. 3 through Fig. 5 that while fracture events generally occurred when the ice islands were subjected to more severe atmospheric and oceanic conditions (*e.g.*, high wind and current speed, air and water temperature, wave energy index, and lifetime mean air and water temperature), warm water temperature played the most important role in the large-scale fracture events of Petermann ice islands. The results also showed

that ice islands subjected to high values of daily-average metocean variables (as specified in Table 4), are expected to have a 75% chance of fracturing. The model validation was performed using k-fold cross-validation approach based on the Pareto principle, and it was found that the error between the estimated fracture probabilities from training and test sets ranged from 13% when only water temperature criterion is considered, to 39% for the full set of criteria.

The results of this study provide an important step toward the development of a probabilistic forecast model for

fracture events of ice islands. The model presented here was built on the fracture event data associated with the ice islands that originated from Petermann Glacier, and therefore applies only to specific ice islands which share similar ice strength properties. Ice islands from other glaciers may have higher or lower ice strength characteristics and could experience fracture events under narrower or wider ranges of metocean conditions than presented in this study. The atmospheric and oceanic conditions, and their corresponding fracture event probabilities presented in Table 4 and Table 5 need to be updated if more deterioration data

become available to improve the model accuracy. The atmospheric and oceanic conditions in the presented model were extracted from reanalysis datasets; however, for the presented model to have an operational forecast use, the metocean conditions and fracture event probabilities should be estimated using the inputs from deterministic models that have the ability to provide short-term forecasts. The results of this study can be used with a limited number of variables. For example, in case only daily-average wind and air/water temperature data for the ice islands are available, the model should be restricted to

criteria sets i=1–3. Once more variable data become available (*e.g.*, wave and lifetime mean variables), then the probabilistic fracture event estimations may take into account the expanded criteria sets (*e.g.*, i=4-5).

Future work should focus on improving this model through expanding the deterioration database of Petermann ice islands, as well as evaluating the fracture event data associated with other ice islands. The presented model was developed based on the data from 328 fracture events; however, more data are needed to train and test this model. The probabilistic model

presented here only considered two states for each variable to avoid model saturation given the limited number of data points. If more data become available, one can improve the model resolution by using a greater number of variable states (*e.g.*, three or four). While this study used seven input variables to develop a probabilistic fracture model, future research can also investigate the role of other variables such as sea ice concentration/thickness and ice island size on fracture events of ice islands. Also, the incorporation of in-situ measured metocean data can contribute significantly to further validation of the

presented ice island fracture model. Finally, for this model to have a forecast capability for future ice island fracture events, the presented fracture model needs to be coupled with a drift model able to reliably forecast the positional data. The output of the presented fracture model can generate fracture probability distributions over the forecast drift trajectory from an ice island drift model, which could serve as a framework to predict the most likely locations/times for fracture event occurrence. This would need to be coupled with a size distribution model to estimate the resulting mass of the ice island fragment(s) following

a fracture event. The mass estimation would then need to be incorporated into a drift forecasting model until a fracture event is predicted, when the scheme iterates again. A probabilistic drift model for the Petermann ice islands in the CI2D3 database is currently under development by the same authors, which will then be integrated into the presented fracture model to ultimately present a coupled ice island drift and deterioration forecast model.

The research presented here fills some of the critical knowledge gaps in glacial ice deterioration forecasts. The results

of this study provide an important step in characterizing the atmospheric and oceanic conditions that govern the large-scale fracture events of ice islands, which are important for improving the calibration of operational ice dynamics models. The increase in the air and water temperatures due to the climate change is expected to drive more frequent massive calving events of Petermann Glacier in the future (Münchow et al., 2016), which could lead to the generation of numerous drifting ice islands

off the east coast of Canada. The ability to predict fracture events of these ice islands could contribute to the development of more reliable strategies to mitigate the risks associated with the presence of glacial ice features, which is necessary for supporting safe offshore operations and marine activities in the ice-prone waters off the east coast of Canada.

## 6 Appendix A

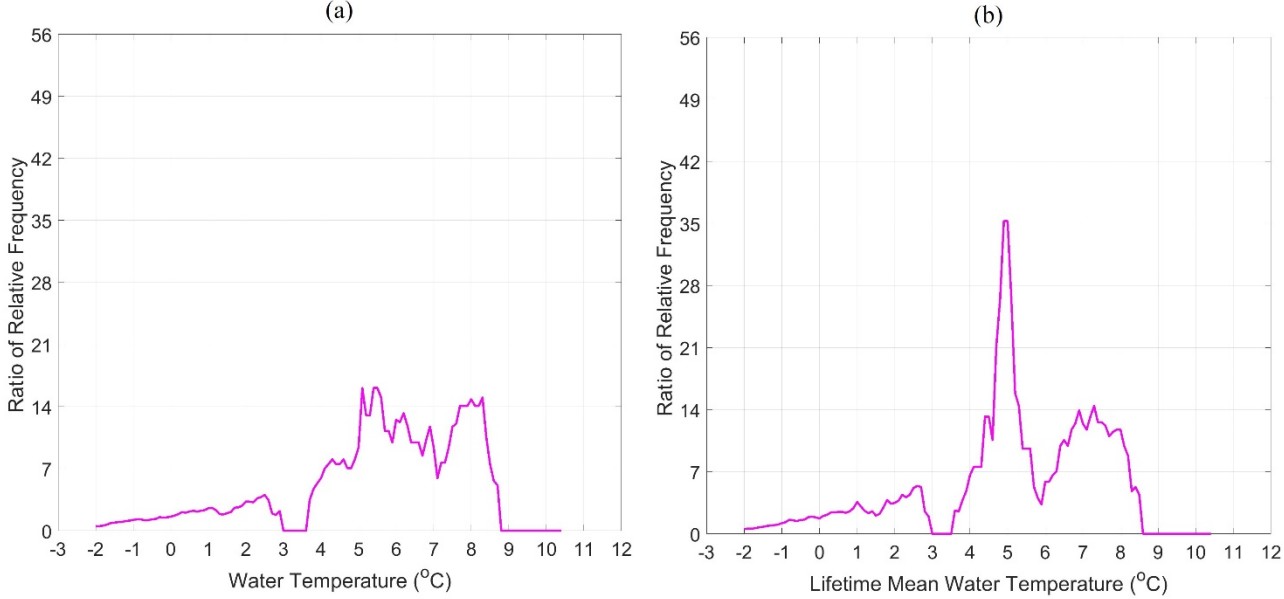

**Figure A1.** The ratio of the relative frequency for fracture events and all observations over the range of water temperature (a) and lifetime mean water temperature (b) that surrounded Petermann ice islands. Values close to one implies that fracture events are as likely to occur as the frequency of observations. Values large compared to one indicates that fracture events are more likely to occur than the frequency of observations. Values less than one implies that fracture events are less likely to occur relative to the frequency of observations.

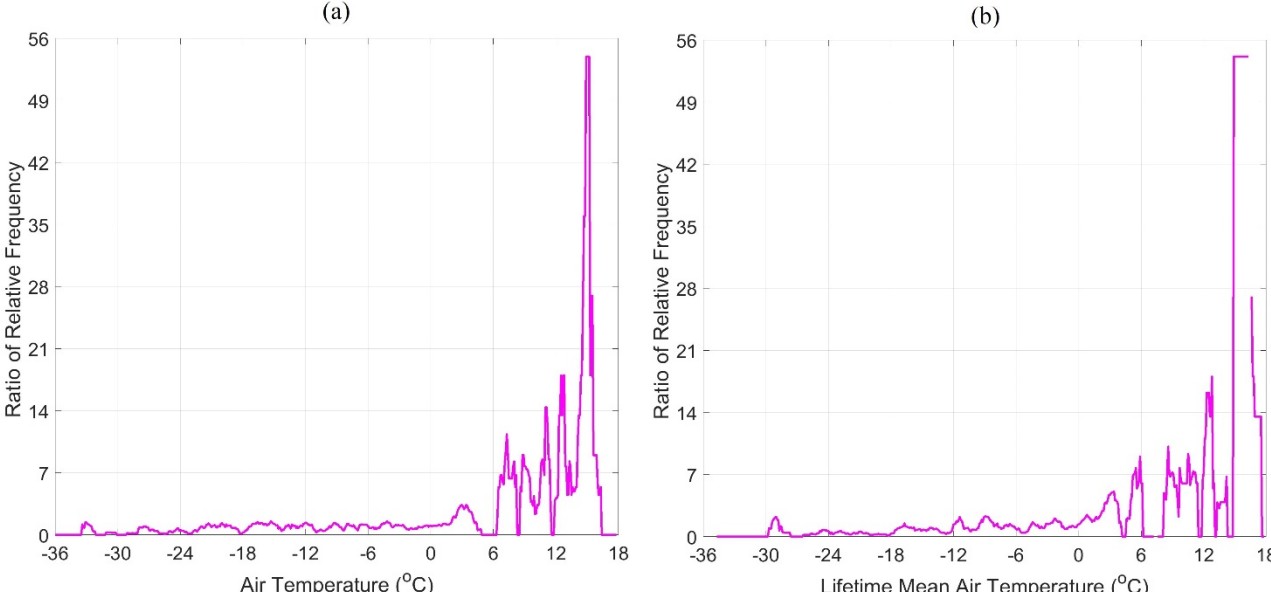

**Figure A2.** The ratio of the relative frequency for fracture events and all observations over the range of air temperature (a) and lifetime mean air temperature (b) that surrounded Petermann ice islands. Values close to one implies that fracture events are as likely to occur as the frequency of observations. Values large compared to one indicates that fracture events are more likely to occur than the frequency of observations. Values less than one implies that fracture events are less likely to occur relative to the frequency of observations.

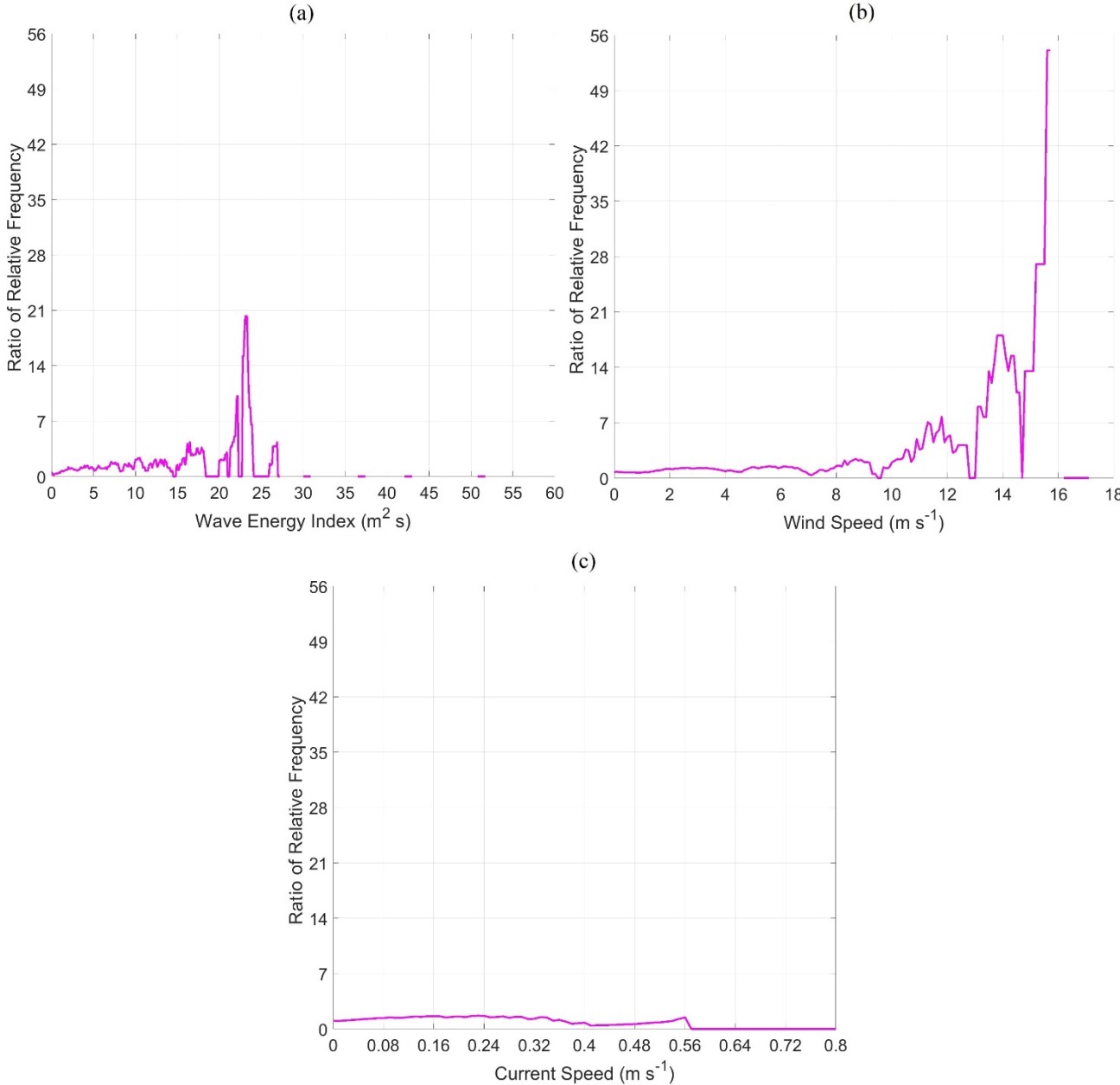


**Figure A3.** The ratio of the relative frequency for fracture events and all observations over the range of wave energy index (a), wind speed (b), and current speed (c) that surrounded Petermann ice islands. Values close to one implies that fracture events are as likely to occur as the frequency of observations. Values large compared to one indicates that fracture events are more likely to occur than the frequency of observations. Values less than one implies that fracture events are less likely to

occur relative to the frequency of observations.

*Data availability.* The CI2D3 database and its documentation can be accessed at http://dx.doi.org/10.21963/12678.

*Author contributions.* IT and RT assisted in the acquisition of the financial support for this study. RZ carried out the reanalysis and CI2D3 data extraction/interpolation with assistance from IT and DM, respectively. RZ developed the methodology with guidance from IT and RT, and analyzed data, developed the model, and prepared the manuscript, with critical review, feedback, and commentary from IT, RT, and DM.

*Competing interests.* The authors declare that they have no conflict of interest.

*Acknowledgements.* The CI2D3 database used in this study was originally developed through a collaboration between the Water and Ice Research Lab (WIRL) at Carleton University and the Canadian Ice Service (CIS). We thank the team of researchers at WIRL, NARR, CMEMS, and ECMWF who provided the database and reanalysis data for this study. We also acknowledge the CIS for providing the digital daily ice charts analyzed in this study. Financial support from Hibernia Management and Development Company (HMDC), MITACS Accelerate Program, and the School of Graduate Studies at Memorial University of Newfoundland is gratefully acknowledged.

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
