# Peer review of "A probabilistic model for fracture events of Petermann ice islands under the influence of atmospheric and oceanic conditions"

_The Cryosphere, 2021_

## Author Response (AR1)

**Manuscript # tc-2021-83 by Reza Zeinali-Torbati, Ian D. Turnbull, Rocky S. Taylor, Derek Mueller: "A probabilistic model for fracture events of Petermann ice islands under the influence of atmospheric and oceanic conditions"**

Dear Editor and Reviewers,

We would like to thank you for your time reviewing our manuscript and providing your insightful feedback. We have refined our paper under your guidance and addressed your recommendations in our manuscript, which significantly improved the quality of our paper.

Here we provide a table of responses that includes our point-by-point response to the corrections/recommendations from the editor (pages. 2-5), reviewer 1 (pages 6-29), and reviewer 2 (pages 30-35). Please note that the line numbers in our responses (right column) refer to the lines on the attached marked-up manuscript when "All Markup" option is selected in Word "Review" tab. However, we were only allowed to upload pdf attachments for the marked-up version. So, please see our supplement zip file for the Word version of our marked-up manuscript.

Thanks again for your insightful review.

| Editor's Comments/suggestions | Authors' Responses |
|---|---|
| **1)** The manuscript presents a probabilistic model for fracture events of Petermann ice lands. My impression after Access Review is that the clarity of the study's scope, the general methods employed meet the required initial quality level. Therefore, the paper is within the scope of TC and I am happy to send it out for peer-review. | Thank you very much for your thorough review and informed comments. It is a great pleasure to see your positive feedback on our paper. Please find below a complete response regarding your comments and suggestions. |
| **2)** Some more detailed remarks from my side (that can be addressed during the revision process) are that:

The distinction between correlation and causation should be addressed more clearly as the model in my impression does not take into account that the fracture events occur on icebergs of different size/thickness/weakness and that there is a potential strong relation with for example latitude. In my impression the paper could gain on making these relations more clear and/or focus more on the aspect of causation than just correlation. | We thank the editor for this helpful comment. In the light of your suggestion, we have added a paragraph in Sect. 3.2 (*lines 415-425*) to address the aspects of causation in ice island fracture events.

As for consideration of other variables and given the limited number of fracture events in our database, we had restrictions on adding more variables to the fracture model. Initially, 10 atmospheric and oceanic variables were considered. However, with the restrictions noted above, it was important to reduce the number of variables. So, the number of variables were reduced to avoid model saturation, which left no room for explicit examination of other variables such as ice island size, thickness, and latitude. Also, we did not have any observed/measured values for the thickness of ice islands. In terms of latitude, while it has not been explicitly considered as an input variable to our model, we can see from the locations of fracture events (Fig. 2-b) that |

| | fracture events are distributed in the study area from northwest of Greenland down to offshore Newfoundland, and not concentrated around a specific latitude. So, we expect this may not play a significant role in fracture events. We should, however, note here that while ice latitude is not considered in the presented fracture model, it is explicitly considered as a proxy for the Coriolis force in a probabilistic drift model, which is currently under development by the same authors. The upcoming drift model aims to build on the presented fracture model to provide a framework for a coupled fracture/drift model for future prediction of ice island fracture events. |
|---|---|
| **3)** The result sections still contain large parts that should belong to the method section (e.g. L350-364 + beginning of section 4). It would be beneficial to move these parts to the method section. | We have revised these sections accordingly and moved the parts describing the methodology to Sect. 2.3 (*lines 286-295*). |
| **4)** It would be beneficial to have a separate discussion section | We appreciate your suggestion and understand that separate results and discussions sections might be helpful in some manuscripts, but we believe our results and discussions are quite linked to each other in a way that splitting in the main section of our paper could cause disconnection between the main points of our study. We think presenting our results and discussions in one section provides a more clear and concise |

| | presentations of our results, which makes it easier for the readers to follow the points discussed in our paper and save them the time they would have otherwise spent on switching between sections. |
|---|---|
| **5)** It is not clear where the thresholds in Table 4+5 originate from | The selected threshold ($x^*$) for the variables in Table 4 were identified by varying each criterion over the range of each variable from the fracture subset to maximize the fracture event probability. The criteria sets in Table 4 were selected by the progressive addition of one or two conditions to the previous criteria set, so that the associated conditional fracture probability would increase. These sentences have already been included in our manuscript (*lines 259-261* and *448-450*).

The thresholds in Table 5 represent the median values of the model variables (as presented in Figs. 3-5). So, we have added a sentence to clarify where those thresholds originate from (*line 619*). |
| **6)** Lat/Lon maps are much distorted over polar regions and it would be beneficial to remap them in a coordinate system that shows less distortion. | We thank the editor for bringing this to our attention. We have revised our maps (Figs. 2 and 6) accordingly to represent the appropriate projection. |
| **7)** However, it will be the task of the reviewers to assess whether these remarks require attention with regard to the conclusions drawn in the paper. They could easily be considered in the revision after the discussion stage. | We have addressed all the commends from the reviewers and revised our paper accordingly. Please find attached our complete responses |

| | regarding the comments and suggestions from the reviewers. |
|---|---|

| Reviewer 1's Comments/suggestions | Authors' Responses |
|---|---|
| **1)** The present study tackles an important problem not only for real-world applications and offshore operations, but also for numerical modelling of icebergs. While there is some knowledge about melting and wave erosion of icebergs, the fracturing of icebergs is a process that is not well understood and therefore still usually missing from models, and only a handful of studies have mentioned or even tackled this issue. Zeinali-Torbati et al. present a timely paper with a probabilistic fracture model for ice islands as a function of the underlying oceanic and atmospheric conditions that could be of high interest to marine offshore activities in the Canadian Arctic, and conceptually it is also very interesting for the inclusion in general iceberg forecasting models.

The paper is generally well-written and presented in an understandable manner. The quality of the figures is okay. I think that the authors address a topic that is of considerable interest and there are only very few papers about that topic so far, so I would like to see the study published. | Thank you very much for your thorough review and informed comments. We also greatly appreciate your acknowledgement of our work's potential for inclusion in iceberg forecasting models and contribution to marine offshore activities in the Canadian Arctic. It is a great pleasure to see your positive feedback on our paper and your interest to see our paper published in The Cryosphere. Please find below a complete response regarding your comments and suggestions. |
| **2)** Specifically, however, there are two studies that go into a very similar direction and that are not discussed. First of all, this is the 3yr-old study by Bouhier et al. (2018) that was published in the same journal (The Cryosphere), and secondly the high-impact study by England, Wagner and Eisenman (2020) in Science Advances. In my opinion, it is import that these two studies are appropriately discussed and cited. | We thank the reviewer for bringing these papers to our attention. These papers are certainly relevant and important to discuss in our manuscript. These papers are now cited and discussed throughout our paper, which have significantly strengthened our paper's relevance to previous research. The following texts were added to the relevant paragraphs, and citations were made to Bouhier et al 2018 and England et al 2020 elsewhere 11 and 5 times, respectively: |

[revised manuscript text omitted]

106/107 "To date, there is no deterministic model to describe the large-scale fracture mechanisms as a function of the metocean conditions that govern these events" | We have incorporated the reviewer's suggested edit and revised some statements in our paper. The sentence now reads:

"**To date, there are only a few deterministic models to describe the large-scale fracture mechanisms for icebergs (*e.g.*, Diemand et al., 1987; Wagner et al., 2014; Bouhier et al., 2018; England et al., 2020).**" (*lines 114-115*). |

| | |
|---|---|
| **4)** 113/114 "However, these models did not account for the relative role of metocean conditions in the fracture processes." | This statement refers to the numerical models cited in our paper, which, in fact, did not consider the impact of environmental conditions on fracture. To clarify this, the sentence was revised as follows: "**However, these numerical models did not account for the relative role of metocean conditions in the fracture processes.**" (*lines 123-124*). |
| **5)** 448 "Therefore, it is impossible to compare the methodologies and results of the presented Bayesian fracture model with an existing physical ice island fracture model" | As per your recommendation, we toned down this statement according to the papers that we added. We also added a sentence before this statement to highlight the difference between the model by Bouhier et al. (2018) and our fracture model. This sentence has been revised and now reads: "**While the bulk volume loss model by Bouhier et al. (2018) can provide a representation of iceberg relative volume loss variation with sea surface temperature and iceberg velocity, it is not able to estimate iceberg fracture probability under the influence of environmental variables presented here. Therefore, it is difficult to provide a validation scheme that can appropriately compare the results of the presented Bayesian fracture model with an existing**|

| | |
|---|---|
| | **physical ice island fracture model.**" (*lines 541-545*). |
| **6)** Bouhier et al., in their section 5, analyse different environmental parameters (SST, current speed, relative velocity between iceberg and currents, wave height, wave peak frequency, wave energy). They note that fragmentation is a complex process and due to its stochastic nature, "individual calving events cannot be forecast. Yet, fragmentation can still be studied in terms of a probability distribution of a calving". They conclude that the highest correlations are found for the ocean temperature (and the iceberg velocity) while the wave-related quantities show no significant link with the volume loss. Ultimately, they present a simple (deterministic) bulk model based on some environmental parameters that somewhat mimics the effect of the fragmentation of large icebergs, and that could -to my understanding- serve as a comparision/benchmark for your work. | We appreciate the explanation that is provided on the fracture model by Bouhier et al. (2018). In the light of your comment, we have added several statements to our manuscript to discuss how the methodologies and results of our work compare against the fracture model presented by Bouhier et al. (2018). These additions can be found in our response to your comment # 2, which we believe have improved the quality of our paper. |
| **7)** 446 "To date, no probabilistic or deterministic models have been presented to investigate the atmospheric and oceanic conditions that lead to the highest probability of large-scale fracture event occurrence for ice islands." | This sentence has been toned down and revised as follows: "**To date, there has been limited research (*e.g.*, Bouhier et al., 2018) investigating the atmospheric and oceanic conditions that lead to the highest probability of large-scale iceberg fracture event occurrence.**" (*lines 540-541*). |
| **8)** 115-117: "To date, no previous research has adopted probabilistic methods (e.g. Bayesian approach)" While this might be true for the "Bayesian approach", England et al. add a stochastic/probabilistic representation of the "footloose mechanism" (cited in your paper) | You are correct, the paper by England et al. (2020) presents a stochastic/probabilistic approach for modeling iceberg fractures. This paper has been cited and discussed throughout our manuscript, as follows: |

into an iceberg drift and decay model, with clear success (their Figures 3 and 4). They note, however, that the breakup scheme is still relatively idealized and based on assumptions. For example, in their study the probability of a child iceberg breaking from the parent iceberg is set as constant in time, while it should depend on SST, sea ice, the roughness of the sea etc.

"**England et al. (2020) presented an approach for modelling the fracture events of large tabular icebergs by incorporating a stochastic representation of the "footloose mechanism" (Wagner et al., 2014) into the analytical iceberg drift by Wagner et al. (2017). The authors showed that coupling their fracture model with an analytical drift model significantly impacted the iceberg meltwater distribution and resulted in improved simulated iceberg trajectories. England et al. (2020), however, noted that the fracture mechanism in their model is simplified based on several assumptions, a key one being the probability of a child iceberg fracturing from the parent iceberg is set as constant in time. However, this parameter should be, in fact, dependent on the environmental variables such as sea surface temperature.**" (*lines 133-139*).

"**Sea ice, however, may play a role in fracture events of ice islands in other regions (*e.g.*, England et al., 2020), so the presented model would need to be extended for application in such regions.**" (*lines 232-233*).

"**Warm surface waters also plays an important role in the initiation of fractures**

**on large tabular Antarctic icebergs (England et al., 2020) through edge-wasting (*c.f.*, Scambos et al., 2005).**" (*lines 344-345*).

We also revised our statement accordingly, which now reads:

"**To date, no previous research has adopted Bayesian approach to predict the probability of ice island fracture events under the influence of the metocean conditions that control these events, likely due to the lack of reliable data.**" (*lines 141-143*).

**9)** The present paper will be much more compelling if the relationship to this previous work is appropriately discussed (in terms of advantages, disadvantages, similarities).

Alternatively, going even further than that, a version of the bulk formula by Bouhier et al. could in principle be used as a comparison, or you could discuss how to better choose the probability of a child iceberg breaking from the parent, which was chosen as a constant in time by England et al.. These latter changes would require work but are however not urgently needed for the present study, in my opinion.

We have added several statements in our paper to discuss the differences between the methodologies, as well as the similarities between the results of our paper, in comparison to the previous fracture models you noted (Bouhier et al., 2018; England et al., 2020). Please find our corresponding added texts in our response to your comment # 2 above.

It would also be certainly interesting to see how the results of our fracture model can be validated against the fracture models by Bouhier et al. (2018) and England et al. (2020). However, the fundamental differences between the assumptions and methodologies for these papers make it very

challenging, and we find this would be beyond the scope of our paper. For example, England et al. (2020) noted that the fracture mechanism in their model is relatively simplified based on several assumptions, a key one being the approximation that the probability of a child iceberg fracturing from the parent iceberg is set as constant in time. However, this parameter should be, in fact, dependent on the environmental variables, which we addressed in our paper. Also, while the bulk volume loss model by Bouhier et al. (2018) can provide a representation of iceberg relative volume loss variation with sea surface temperature and iceberg velocity, it is not able to estimate iceberg fracture probability under the influence of environmental variables presented here. Therefore, it is difficult to provide a validation scheme that can appropriately compare the results of the presented Bayesian fracture model with the two fracture models noted above.

| | |
|---|---|
| **10)** Another slight weakness of the paper, as far as I understand it, is that you are considering only 328 fracture event days. If I understand correctly, you do not allow for a shift around that date. So any extreme conditions (in air temperature for example), even a single day before the calving event, are potentially missed and can only enter your model via the "lifetime" air temperature? Depending on the length of the iceberg's life, | This is certainly an interesting point, which has already been investigated during our model development, but not presented in our paper. We previously tested the model performance with two-week mean values for air and water temperatures, however, they |

which can be months up to years, I am worried that short-lived extremes could thus be rather hidden in this long-term mean for the oldest bergs.

Instead of lifetime air temperature, average temperatures for the previous 7-day (or 14-day?) period might help in that regard. Furthermore, I have the impression that example timeseries for the days around fracture events (for your considered 7 main variables in Table 1) could help the reader to understand your choices better and illustrate the likely major (minor) role of some of them in causing iceberg fracture.

did not make significant difference in the outcome and brought no improvement.

Given the significant role of warm waters in iceberg deterioration (Kubat et al., 2007; Bouhier et al., 2018), we expected a link between the ice island fracture events and the cumulative effect of the temperature variables over their lifetimes, which cannot be captured using the short period (*e.g.*, 7-day or 14-day) mean values. Our analysis of the lifetime temperature values presented in Figs. 3 and 4, in fact, shows a link between the lifetime mean values and the occurrence of the fracture events. To address your comment and better clarify this in our paper, however, a sentence has been added, explaining that short period mean temperature values were assessed, but did not improve the model performance. The newly added texts read:

"**To better capture the short-lived extreme conditions in air and water temperatures prior to fracture events, the two-week mean values for air and water temperatures were also tested, but they did not make significant difference in the outcome and brought no improvement to the model performance, so they were subsequently excluded from the model inputs.**" (*lines 219-222*).

| | A few sentences have been added to explain how our presented ice island fracture model can be coupled with a drift forecasting model. The added texts read: |
|---|---|
| **11)** Another suggestion would be to say some words in the Discussion about how you plan to add this model for fracture events to an iceberg drift (forecasting) model? (l. 519, l. 532-534) | "**The output of the presented fracture model can generate fracture probability distributions over the forecast drift trajectory from an ice island drift model, which could serve as a framework to predict the most likely locations/times for fracture event occurrence. This would need to be coupled with a size distribution model to estimate the resulting mass of the ice island fragment(s) following a fracture event. The mass estimation would then need to be incorporated into a drift forecasting model until a fracture event is predicted, when the scheme iterates again.**" (*lines 646-651*). |
| **12)** Say you determine a probability of 28% in the field, given your environmental conditions (as in l. 358-359). If your berg is still intact the next day under similar environmental conditions, how does this change the probability for fracture? What if you begin to check hourly, does this change the probability? I am probably wondering about the time-step dependence of your model (see also equation 4 in England et al.). If you can give some hints for what you are considering in your iceberg forecasting model that is in development, this would be greatly appreciated. | This is a very interesting point to assess how fracture probabilities are impacted if we consider the historical probability estimations over the drift path of the given ice island. This can be investigated by including an input variable that accounts for the sum of fracture probabilities over ice island's lifetime. However, with the limited number of fracture events in the CI2D3 database, there is insufficient data to allow |

| | for such an addition to our model inputs as it would increase the number of state combinations, and consequently results in the saturation of our model. We should acknowledge here that our model is able to predict fracture probabilities based on the given daily-average and lifetime mean variables, which is independent of the fracture event probability in the previous day. In the case of your example, if the environmental conditions for the next day remain the same, then the model would predict the same fracture probability. The cumulative effects only enter the model via the lifetime mean variables. While the daily rate for the number of fractured icebergs (r) in Eq. 4 of England et al. (2020) is assumed to be constant, our model fracture rate is not constant in time and changes with the metocean conditions. Our model time-step is daily (see section 2.2), so the environmental conditions can be updated on a daily basis to present daily fracture probability estimations. |
|---|---|
| **13)** A last question in that regard is the following: Imagine you have hundreds of icebergs drifting through similar environmental conditions, what is the "expected number of days" an iceberg can drift through a 75% fracture corridor zone?

In l.419 you state that one ice island drifted for about 14 days in the medium-high fracture | We appreciate the point and explanation that is provided. It is certainly interesting to investigate the number of days an ice island can survive (from fracturing) in a given fracture corridor zone, which requires the addition of an input variable such as the |

| | |
|---|---|
| probability zone. Intuitively, this seems rather unlikely, given that every day for two weeks the ice island was apparently more likely to fracture than to stay intact. Since we are dealing with probabilities, however, also unlikely trajectories can happen in reality. So could one maybe compute a theoretical upper limit of days of survival - and was this ice island close to it? | "cumulative fracture probability" (*i.e.*, the sum of fracture probabilities over ice island's lifetime or a certain period) to our model. However, given the limited number of fracture events in the CI2D3 database, we have insufficient data that restricts the number of input variables (and their state combinations) that can be used in the presented model, otherwise our model would be saturated.

It is also worth noting that the discussion you noted refers to the case study presented in Fig. 6-a, where the fracture probabilities are much lower than the 75% probabilities that are associated with extreme metocean conditions in Table 4. We acknowledge that the term "Highest fracture probability" that we used in Fig. 6 might cause confusion, so, we have removed the terms "Lowest", "Medium", and "Highest" from the legends in Fig. 6 to better clarify the distinction between the case note above. |
| **14)** Abstract: „presented" -> present tense maybe? | The verbs in the "Abstract" section have been changed to present tense. |
| **15)** „Bayes theorem" -> „Bayes' theorem" | The term "Bayes theorem" has been changed to "Bayes' theorem" throughout the manuscript (*lines 17, 480*). |

| | |
|---|---|
| **16)** l.75 surface area "of" ice islands | The preposition "of" has been added to the sentence (*line 75*). |
| **17)** l.96 originated from "the" 2012 calving event | The article "the" has been added to the sentence (*line 100*). |
| **18)** l.104 "convection caused by iceberg rolling" Is there a citation for this? | A citation has been added to the sentence. The concerned sentence now reads:

"**However, there are other mechanisms associated with iceberg deterioration such as large-scale fracture caused by internal stress and convection caused by iceberg rolling (Kubat et al., 2007).**" (*lines 107-109*). |
| **19)** 129 Barbat et al. (2019) also find a power law distribution for Antarctic near-coastal icebergs (their Fig. 5), https://doi.org/10.1029/2019JC015205 | The study by Barbat et al. (2019), as well as the one by Bouhier et al. (2018) have been cited in the sentence and added to the reference list. The updated sentence now reads:

"The authors also revealed that fracture processes significantly contributed to the overall deterioration of Petermann ice islands as the ice island size distribution followed a power law model, which was corroborated by the results of Stern et al. (2016), Tournadre et al., (2016), and Enderlin et al. (2016)**, Bouhier et al. (2018), and Barbat et al. (2019)**" (*lines 156*). |

| | |
|---|---|
| **20)** 145 , l. 256 Most often you refer to the "parent-child" relationship, sometimes to "mother-daughter". Maybe you could use the former more consistently | This change has been made throughout the manuscript (*line 316*). |
| **21)** 149 Since the total lifespan is differently long for different icebergs, have you considered something like 7-day running means instead of "lifetime" ("the week before potential fracture")? (where "7 "can be replaced by any number that sounds reasonable to you) | This has already been addressed, please see our response to your comment # 10 above. |
| **22)** 175 Did you ever consider something very simple like "latitude"? | We have not explicitly considered "latitude" as an input variable to our model. However, from the locations of fracture events presented in Fig. 2-b, we can see that fracture events are distributed in the study area from northwest of Greenland down to offshore Newfoundland, and not concentrated around a specific latitude. So, we expect this may not play a significant role in fracture events. Also, given the limited number of fracture events in the database we used, we had restrictions on adding more variables to the fracture model. Initially, 10 atmospheric and oceanic variables were considered. However, with the restrictions noted above, it was important to reduce the number of variables. So, the number of variables were reduced to avoid model saturation, which left no room for explicit examination of other variables such as ice latitude. |

| | We should, however, note here that while ice latitude is not considered in the presented fracture model, it is explicitly considered as a proxy for the Coriolis force in a probabilistic drift model, which is currently under development by the same authors. The upcoming drift model aims to build on the presented fracture model to provide a framework for a coupled fracture/drift model for future prediction of ice island fracture events. |
|---|---|
| **23)** 190 You mean you normalized by the number of days the ice island drifted? (because dividing by the timespan in seconds would result in a weird unit)

189-191 I think this is described in a very complicated manner. Don't you just take the mean of the daily values? | You are correct, this sentence has been revised to clarify. The concerned sentence now reads:

"**These daily-average values were then averaged over the number of days the ice island drifted to effectively compute the lifetime mean wave energy index, as well as the mean air and water temperatures over the lifespan of the ice island, which differ from positive degree day calculations that are often used in ice melt rate models (*e.g.*, Hock, 2003).**" (*lines 217-219*). |
| **24)** 198 This could be a good line to mention that sea ice will play a role in other conditions or regions on Earth (e.g. England et al. 2020), so that your model would need to be extended for other applications | A sentence has been added to mention the potential role of sea ice in fracture events of ice islands in other regions. The added sentence reads: |

| | "**Sea ice, however, may play a role in fracture events of ice islands in other regions (e.g., England et al., 2020), so the presented model would need to be extended for application in such regions.**" (*lines 232-233*). |
|---|---|
| **25)** Figure 1: How do you determine the direction of the causality? Why does air temperature "cause" water temperature and not vice versa (they are tightly coupled) | You are right, the air and water temperature are tightly coupled. Fig. 1 has been revised to represent this two-way causality. |
| **26)** 207/208 Is this due to the (relative to CMEMS) lower spatial resolution of ERA-Interim? Are the ice islands coinciding with land boxes then? | Yes, the ERA-Interim data have a lower spatial resolution (1/8°) than CMEMS data (1/12°). The lower number of available data for waves is likely due to the fact that the ice islands drifted some time near coastlines, and the extracted data have insufficient spatial resolution to model data close to the coastlines. |
| **27)** 218/219 Again, how do you decide on causality between, e.g., air and water temperature? Also, it would be great to add the r values to the Figure. | As per your comment above, Fig. 1 has been revised to include the two-way causality between air and water temperatures. Also, the correlation coefficient values (r) for the given correlations have been added to Fig. 1. |
| **28)** 220 "high metocean conditions" -> maybe "extreme metocean conditions" | The term "extremely high metocean conditions" has been changed to "extreme metocean conditions" (*line 257*). |
| **29)** 1.223/224 No brackets around the reference | The brackets have been removed (*lines 262*). |

| | |
|---|---|
| **30)** Table 1: The notation is not clear to me. Do you subtract the median value, or is V_(w-x) just the median of all V_w values? Could you give numbers here as well? | The notation $\tilde{x}$ represents the median value in the distribution of the given variable. To better clarify this, the notation "$-\tilde{x}$" has been changed to "$,\tilde{x}$". The median values are not presented in Table 1, given that this table is in the Methodology section, and the median values are a part of the Results section. The median values, however, were presented in Figs. 3-5. |
| **31)** 280 "This indicates the important contribution of warm waters to faster deterioration of glacial ice features…" See also papers mentioned above, where SST is considered | A few sentences have been added to include a citation for these papers. The added sentences read:

"**The significant contribution of water temperature to fracturing process was corroborated by Bouhier et al. (2018), where a significant correlation between iceberg relative volume loss and sea surface temperature was found. Warm surface waters also plays an important role in the initiation of fractures on large tabular Antarctic icebergs (England et al., 2020) through edge-wasting (*c.f.*, Scambos et al., 2005).**" (*lines 342-345*). |
| **32)** Figure 3 and even more so, Figure 4, shows strong signs of bimodality. Is that why you split into two states in Table 1? This is unclear.

Furthermore, an immediate question is whether the two modes (in Figure 4) are potentially due to different seasons, or whether the fracture events for the two modes are maybe spatially | The variables were split into two states based on the observed median value for each variable (Table 1). The bimodality in Figs. 3 and 4 is due to the ice islands' drifts in different seasons. Our analysis tracked Petermann ice islands from July 2008 to |

| | |
|---|---|
| clustered in specific areas? This could also potentially hint at different mechanisms involved, which is very difficult to assess from the histograms alone. | December 2013, so the ice islands experienced various air and water temperatures over their lifespans. Our preliminary analysis of the locations of the fracture events (Fig. 2-b) shows no specific cluster as the fracture events were distributed over the study area from the northwest of Greenland down to offshore Newfoundland. |
| **33)** 314 I was wondering whether instead of the mean (wave energy index), the maximum during a day could be more telling (I do not know whether that is available in the reanalysis). Same for winds etc | It is certainly an interesting point. As per your comment, we looked into the reanalysis datasets that we used, however, the daily maximum values for these variables are not available in the datasets. |
| **34)** Figure 3 and 4: Could you add the median line for all observations in the right panels so that one can see the displacement for the median of the fracture events directly?

In general, are the different medians significantly different from each other in a statistical sense (see e.g. l. 322 "slightly greater")? | Figs. 3 and 4 have been revised to include the median values from all observations (a) on the right panel (b).

As for the medians, we have conducted a Mann–Whitney U test and found out that the variable medians for the fracture events and all observations are statistically different. So, we have replaced the term "slightly" with "statistically" to clarify (*lines 375, 388*). |
| **35)** Figure 5: Please add more ticks; add median line for all observations in the right panels. Figure 5 caption: "for a) all observations (n=3985) and for b) n=131 fracture events" | Your suggested edits have been incorporated and the adjustments have been made on Fig. 5. |

| | |
|---|---|
| **36)** 335: This is a much clearer definition for the lifetime mean variables that could be given in the beginning of the paper | A clearer description of the lifetime mean variables has been added in the beginning of the paper. The added sentence now reads:

 "**These daily-average values were then averaged over the number of days the ice island drifted to effectively compute the lifetime mean wave energy index, as well as the mean air and water temperatures over the lifespan of the ice island, which differ from positive degree day calculations that are often used in ice melt rate models (*e.g.*, Hock, 2003).**" (*lines 217-219*). |
| **37)** 346 No bracket in the end | The bracket have been removed. |
| **38)** Table 4: Where do the numbers/thresholds come from in this table? Are these the V_w-x values from Table 1? I might have missed that part in the paper | The selected threshold in Table 4 were identified by varying each criterion over the range of each variable from the fracture subset to maximize the fracture event probability. This was mentioned in section 2.3. |
| **39)** 370 "the addition of the lifetime mean variables did not increase the fracture probability above 75%" See my previous comments on whether the previous 7-to-14-day-means before determining the probability could be more telling than "lifetime" values | This has already been addressed, please see our response to your comment #10 above. |
| **40)** 380 Maybe also different variables might need to be considered (sea ice)? | A sentence has been added at the end of the paragraph to explain this. The added sentence read: |

| | |
|---|---|
| | "**Under such conditions, the model variables themselves could also be modified if additional variables (*e.g.*, sea ice concentration) were deemed to be important.**" (*lines 470-472*). |
| **41)** 400 I think you could start another subsection here, e.g. "3.4 Case study" | A subsection has been added and titled "**3.4 Case study**" (*line 492*). |
| **42)** 422 "towards the end of its drift period off Labrador coast" -> "towards the end of its hypothetical drift off Labrador coast, which could thus likely have been longer than the 2010-2011 drift." | This change has been applied (*lines 514-515*). |
| **43)** 448 "Therefore, it is impossible to compare the methodologies and results of the presented Bayesian fracture model with an existing physical ice island fracture model" Given the suggested papers above, I don't think this is entirely accurate. It would certainly require a great deal of work to compare to other approaches in previous papers (that are also tested for the other hemisphere only), but it is not impossible. | We have included your suggested papers above in our manuscript and have adjusted this statement accordingly, which now reads: "**To date, there has been limited research (e.g., Bouhier et al., 2018) investigating the atmospheric and oceanic conditions that lead to the highest probability of large-scale iceberg fracture event occurrence. While the bulk volume loss model by Bouhier et al. (2018) can provide a representation of iceberg relative volume loss variation with sea surface temperature and iceberg velocity, it is not able to estimate iceberg fracture probability under the influence of environmental variables presented here. Therefore, it is difficult to provide a** |

| | validation scheme that can appropriately compare the results of the presented Bayesian fracture model with an existing physical ice island fracture model.” (*lines 540-545*). |
|---|---|
| **44)** 473 fracture probability | “fractures probabilities” has been changed to “fracture probabilities” (*line 553-554*). |
| **45)** 478 the number of … increases | The word “increase” has been changed to “increases” (*line 559*). |
| **46)** 485-488 Could you give a good example for very implausible/unlikely combinations? | An example of a very unlikely combination has been added. The added texts read:

“**As an example, the condition where $T_w > -0.3℃, V_w > 2.8ms^{-1}, T_a \leq -2.1℃), E_w > 5.1m^2s, T_{a_{avg}} > -3.5℃,$ and $T_{w_{avg}} \leq -0.7℃$ was a very unlikely combination that only occurred once among all ice island observations, and no fracture event occurred under such conditions.**” (*lines 601-603*). |
| **47)** 504 SST was also found to be a leading variable in the suggested papers above | Citations of the papers you suggested have been added earlier in the paper to note the important role of SST to iceberg fracturing process. The added texts read:

“**The significant contribution of water temperature to fracturing process was corroborated by Bouhier et al. (2018), where a significant correlation between** |

| | |
|---|---|
| | **iceberg relative volume loss and sea surface temperature was found. Warm surface waters also plays an important role in the initiation of fractures on large tabular Antarctic icebergs (England et al., 2020) through edge-wasting (*c.f.*, Scambos et al., 2005).**" (*lines 342-345*). |

| Reviewer 2's Comments/suggestions | Authors' Responses |
|---|---|
| **1)** This study presents a probabilistic model of iceberg fracture based on a series of ice islands generated from calving events from the Petermann ice tongue with the goal of stepping towards providing a real world practical operational forecast model. The authors analyzed the role of wind speed, air temperature, ocean current speed, water temperature and something called the wave energy index along with mean air temperature and sea ice concentration.

As someone who works largely on the mechanical side I don't have experience with the operational side or the statistical framework. Someone who works more closely on that side of the field will have a better idea of the appropriateness of the methodology and relationship to prior work. Overall, however, I don't see any obvious objections to the statistical tests or procedures used. A minor comment is that it would be helpful to relate the probabilistic model more closely to process level models of iceberg decay, although that may follow in subsequent work. | Thank you very much for your review, feedback, and suggestions. It is a pleasure to see your positive feedback on our paper. As you noted, it would certainly be an interesting topic for a future work to investigate how the presented probabilistic model relates to process level models for iceberg deterioration. Please find below a complete response regarding your comments and suggestions. |
| **2)** Overall, I only have a few minor comments.

1. How reliable are the inputs fed into the model? We are presented with a probabilistic model driven by inputs. Reanalysis and wave forecasts all have strengths, but also uncertainties. Hence the question from a non-expert as to whether the uncertainty in the model inputs small enough to be neglected? | We have looked into the documentations for the reanalysis data that we used to explore if there are any reported error/uncertainty on the extracted data that we used in our study. However, these products have not reported on the errors in the datasets created. We have, however, found a few studies that reported on the accuracy of some ocean products in other regions. Surface currents are expected to be the most uncertain variable in drift forecasting models. The mean error in the speed of surface currents |

| | from CMEMS Global Ocean Physics Reanalysis model was reported to be $0.08 \text{ m s}^{-1}$ (Lellouche et al., 2018). CMEMS water temperature data were reported to be within 1.2 °C of measured data, with RMS error at sea surface being around 0.4 °C (Sukresno et al., 2019). The Root Mean Square Error (RMSE) in significant wave height estimates from ECMWF was reported to be less than 0.37 m (Wang et al., 2019). The mean Recursive Prediction Error (RPE) for wave height and wave period was reported as 12.5% and 7.7%, respectively. Also, a bias of 1.5 °C and $0.16 \text{ m s}^{-1}$, respectively, was noted for air temperature and wind speed from NARR (Boccara et al., 2008). |
| --- | --- |
| | We acknowledge the error in the input data. However, we should note here that given our model setup, in which the level of data is reduced to binomial level (Table 1), we expect the presented model to be less vulnerable to the errors in the input data, unless the values are too close to the median values. |
| | References: |
| | • Boccara, G., Hertzog, A., Basdevant, C., and Vial, F. (2008). Accuracy of NCEP/NCAR reanalyses and |

| | |
|---|---|
| | ECMWF analyses in the lower stratosphere over Antarctica in 2005. Journal of Geophysical Research: Atmospheres, 113(D20). |
| | • Lellouche, J. M., Greiner, E., Le Galloudec, O., Garric, G., Regnier, C., Drevillon, M., ... and Le Traon, P. Y. (2018). Recent updates to the Copernicus Marine Service global ocean monitoring and forecasting real-time 1∕12° high-resolution system. Ocean Science, 14(5), 1093-1126. |
| | • Sukresno, B., Murdimanto, A., Hanintyo, R., Jatisworo, D., and Kusuma, D. W. (2019, March). The use of CMEMS and Argo Float data for Bigeye Tuna fishing ground prediction. In IOP Conference Series: Earth and Environmental Science (Vol. 246, No. 1, p. 012002). IOP Publishing. |
| | • Wang, J., Li, B., Gao, Z., and Wang, J. (2019). Comparison of ECMWF significant wave height forecasts in the China sea with buoy data. Weather and Forecasting, 34(6), 1693-1704. |
| **3)** The analysis considers wave energy, but is it also possible to consider wavelength in addition to amplitude? The wavelength of | We appreciate the point and explanation that you provided. It is certainly interesting to |

ocean swell relative to the flexural wavelength of the ice island could be important in determining if bending stresses are large enough to fracture the island. In fact, modest swell events are sufficient to breakup the sea ice pack when the ocean swell as an appropriate period, but long wavelength swell penetrates the sea ice pack with minimal effect.

investigate the addition of a new variable such as wavelength to our model, but the ECMWF ERA Interim dataset does not report on wavelength values. However, our "wave energy index" variable is dependent on wave period (Eq. 1), a component that is tightly correlated with wavelength. Also, given the limited number of fracture events in the CI2D3 database, we have restrictions on the number of input variables (and their state combinations) that can be used in the presented model. So, we have insufficient data to allow for the addition of a new variable, otherwise our model would be saturated. We also looked into the wave height values to investigate if the ice islands were exposed to exceptionally large weight heights over their lifetime to cause significant bending stress. However, our analysis of the extracted data shows that only ~5% of the wave heights were above 2m, with maximum wave height being ~5m. The wave heights over the drift path of the studied ice islands were mostly less than 2m, which is much smaller than the sail height of large ice features such as the ice islands. Therefore, we expect the waves to have minimal impact on the bending stress exposed the studied ice islands. We, however, acknowledge here that waves could

| | have significant impact on sea ice breakup events driven by bending stress. |
|---|---|
| **4)** Can the authors provide a sentence or two providing the motivation and sensitivity for selecting the prior probability distribution? My own experience with Bayesian analysis is that selecting on appropriate prior can be tricky and, unless there is a large amount of data, the prior can play a role guiding predictions. That is not to say that this is the case here, but a few sentences describing the motivation and sensitivity may be useful. | Given the fairly large amount of data in our database, the prior probability of fracture event occurrence was calculated based on our knowledge on the number of fracture events (328) and the number of all observations (17755), before some evidence (metocean conditions) is taken into account. A sentence has been added to explain how the prior probability was calculated. This added sentence reads: "**Given the large size of the CI2D3 database, the value of $P(X)$ was estimated as the frequency of fracture events (*i.e.*, the number of fracture events divided by the total number of observations) before any criteria set based on metocean conditions was considered.**" (*lines 267-269*). |
| **5)** I had a hard time initially interpreting Figure 3 and others. I think what we are supposed to do is compare the figure on the left with the figure on the right to see the enhancement of fracture events at warm ocean/atmosphere temperatures compared to the frequency of observations of warm ocean/atmosphere temperatures. This is quite convincing after contemplating the figures. I wonder if stepping readers not used to this type of plot through what we are supposed to see would be helpful. Alternatively, would it be more useful/intuitive to plot the ratio of the left and | We thank the reviewer for bringing this point into our attention, which can certainly provide more intuitive representation of the conditions where fracture events are more likely to occur. Therefore, we have added three panel figures in the Appendix section (Figs. A1-A3) to show the ratios of the relative frequency for fracture events and all observations over the range of our variables |

right panels to show the enhancement of fracture events in warmer conditions relative to the occurrence of these conditions? In a plot of this type, values close to one would imply that fracture events are as likely to occur as the frequency of observations. Values large compared to one would indicate that fracture events are more likely to occur than the frequency of observations and values less than one would imply that fracture events are less likely to occur relative to the frequency of observations.

(section 6). We have also added a paragraph in section 3.2 to describe the added figures, which reads:

"**The enhancement of fracture events under the conditions where the ice islands experienced higher values of metocean variables was investigated through ratios of the relative frequency for fracture events and all observations over the range of variables presented in Figs. 3-5. These results are presented in Appendix A (Figs. A1-A3), where values close to one imply that fracture events are as likely to occur as the frequency of observations. Values large compared to one indicate that fracture events are more likely to occur than the frequency of observations. Values less than one imply that fracture events are less likely to occur relative to the frequency of observations. The results in Figs. A1-A3 reveal that the ratio of the relative frequency for fracture events and all observations generally increases with the values of metocean variables, which clearly indicate a tendency for fracture events to occur under more extreme conditions.**" (*lines 396-403*).

**6)** Line 71 extra space in "w ave"—>wave

This error was corrected (*line 71*).